# A Huluwa phosphorylation switch regulates embryonic axis induction

Yao Li[1,6], Yun Yan[1,2,6], Bo Gong [ID][3,6], Qianwen Zheng[1], Haiyan Zhou[1], Jiarui Sun[1], Mingpeng Li[1], Zhao Wang[1], Yaohui Li[1], Yunjing Wan[1], Weixi Chen[1], Shiqian Qi [ID][2], Xianming Mo [ID][4], Anming Meng [ID][5], Bo Xiang[1] & Jing Chen [ID][1] ✉

Embryonic axis formation is essential for patterning and morphogenesis in vertebrates and is tightly regulated by the dorsal organizer. Previously, we demonstrated that maternally derived Huluwa (Hwa) acts as a dorsal determinant, dictating axis formation by activating β-catenin signaling in zebrafish and *Xenopus*. However, the mechanism of activation and fine regulation of the Hwa protein remains unclear. Through candidate screening we identified a mutation at Ser168 in the PPNSP motif of Hwa that dramatically abolishes its axis-inducing activity. Mechanistically, mutating the Ser168 residue reduced its binding affinity to Tankyrase 1/2 and the degradation of the Axin protein, weakening β-catenin signaling activation. We confirmed that Ser168 is phosphorylated and that phosphorylation increases Hwa activity in β-catenin signaling and axis induction. Several kinases including Cdk16, Cdk2, and GSK3β, were found to enhance Ser168 phosphorylation in vitro and in vivo. Both dominant-negative Cdk16 expression and pHwa (Ser168) antibody treatment reduce Hwa function. Lastly, a knock-in allele mutating Ser168 to alanine resulted in embryos lacking body axes, demonstrating that Ser168 is essential to axis formation. In summary, Ser168 acts as a phosphorylation switch in Hwa/ β-catenin signaling for embryonic axis induction, regulated by multiple kinases.

Axis induction, a pivotal event in early embryonic development, guides fate map establishment for subsequent body plans[1]. Evidence suggests that embryonic axis induction commences between the late-blastula and early-gastrula stages. This process hinges on the dorsal organizer in vertebrates, known variously as the embryonic shield in zebrafish, the Spemann-Mangold organizer in *Xenopus*, Hensen's node in chickens, and the primitive streak in mammals[2–6]. Organizers across species form under the strict regulation of conserved signals, notably maternal β-catenin signaling[7–12]. In early embryos, maternal β-catenin protein is ubiquitously expressed but becomes specifically activated in the future dorsal region. This activation induces the expression of organizer-specific genes (e.g., *boz/chd/gsc* in zebrafish and *nodal3.1/ siamois1/gsc* in frogs)[10,13]. For a considerable time, studies have shown that Wnt ligands, crucial extracellular stimulators of canonical Wnt/β-catenin signaling, activate maternal β-catenin[13,14] before the discovery of the Hwa protein.

Hwa, a maternal-effect factor, functions as a dorsal determinant. Its mRNA is deposited at the oocyte's vegetal pole and moves to the

[1]Department of Pediatric Surgery and Laboratory of Pediatric Surgery, West China Hospital, Sichuan University, Chengdu 610041, China. [2]Department of Urology, State Key Laboratory of Biotherapy, West China Hospital, Sichuan University, Chengdu 610041, China. [3]Department of Cell and Developmental Biology, Weill Cornell Medicine, Cornell University, 1300 York Avenue, New York, NY 10065, USA. [4]Department of Pediatric Surgery and Laboratory of Stem Cell Biology, State Key Laboratory of Biotherapy, West China Hospital, Sichuan University, 610041 Chengdu, China. [5]Laboratory of Molecular Developmental Biology, State Key Laboratory of Membrane Biology, Tsinghua-Peking Center for Life Sciences, School of Life Sciences, Tsinghua University, Beijing 100084, China. [6]These authors contributed equally: Yao Li, Yun Yan, Bo Gong. ✉e-mail: jingchen@scu.edu.cn

future dorsal site upon fertilization, activating β-catenin signaling[12,15]. Research indicates that Hwa binds to and supports Tankyrase1/2 (Tnks1/2)-mediated degradation of the Axin protein. This process stabilizes β-catenin independently of Wnt ligands and Lrp5/6 coreceptors. Despite being a recently identified protein with some conserved regions, Hwa's functional domains or motifs remain largely unknown so far. Xuchen Zhu et al. recently reported Hwa's lysosomal degradation pathway in frogs[16], yet the precise control and initiation of Hwa signaling to form a correctly sized single axis are still unknown.

Hwa is a single-pass transmembrane protein, comprising a nonessential extracellular domain for axis induction, a transmembrane domain, and a large, disordered intracellular domain with a conserved PPNSP motif, akin to the PPP(S/T)P motif of the Lrp5/6 receptor (Fig. 1a). Studies show that phosphorylation of PPP(S/T)P is necessary for Lrp5/6 activation[17-22]. The Lrp6 protein, lacking the extracellular domain, acts as a constitutively active form, even with a single PPPSP motif linked to LDLRΔN[20]. It remains unclear if Hwa operates similarly as a constitutively active receptor-like Lrp6ΔN without its extracellular domain. Also, questions arise regarding the regulation of Hwa activity through PPNSP motif phosphorylation and the kinases or regulators involved. Addressing these questions is essential for fully understanding Hwa signaling in embryonic axis induction.

In this study, we conducted candidate sites screening and axis induction experiment using *hwa* maternal mutant (M*hwa^tsu01sm/tsu01sm*) embryos. Our findings highlight the role of Ser168 in the PPNSP motif for Hwa signaling activation. We characterized the phosphorylation state of Ser168 in Hwa in both HEK293T cells and zebrafish embryos. Additionally, using in vitro and in vivo phosphorylation assays, we identified at least three kinases (Cdk16, Cdk2, and GSK3β) that enhance the phosphorylation at Ser168 in Hwa. In zebrafish embryos, Cdk16, Cdk2, and membrane-tagged GSK3β (mGSK3β) amplified the axis-inducing activity of Hwa. Conversely, dominant negative Cdk16 (Cdk16^DN) and pHwa (Ser168) antibody reduced its activity. Furthermore, a knock-in allele generated by CRISPR mutating Ser168 to alanine resulted in embryos lacking body axes, demonstrating that Ser168 is essential to axis formation. These results collectively underscore the importance of Ser168 phosphorylation in activating Hwa/β-catenin signaling for embryonic axis induction, a process modulated by multiple kinases.

## Results

### Mutation of Ser168 eliminates the axis-inducing activity of Hwa protein
Previously, we characterized two conserved motifs in Hwa, PPNSP and RRSST[15], which may contain the essential sites, potentially crucial for Hwa signaling activation. To evaluate this hypothesis, we conducted rescue experiments using point-mutated *hwa* mRNAs in M*hwa^tsu01sm/tsu01sm* embryos. For the RRSST motif (Aa181-190), single-amino acid substitutions weakened the rescue effect. Notably, the last three polar amino acids ([188]SST[190]) significantly affected the activity. However, these sites retained some activity, even with double or triple mutations. For the PPNSP motif, our analysis at 24 hour post fertilization (hpf) revealed that alanine substitution of Ser168 (S168A) almost eliminated the axis-inducing function. Mutations at adjacent sites variably weakened the rescue effect of *hwa* mRNA (Fig. 1b, c). The expression of the organizer marker gene *chd* at 6 hpf further confirmed the loss of axis-inducing activity in the S168A mutation (Fig. 1d). Hwa protein, present in most Chordata except Aves and Mammalia, features a conserved PPN(S/T)P motif, illustrated with a brown shadow (Fig. 1e). Wild-type Hwa from various species (amphioxus, sea squirt and frog) partially rescued axis formation in M*hwa^tsu01sm/tsu01sm* embryos, albeit with a low success rate. However, alanine substitution of the corresponding serine/threonine residue greatly reduced or eliminated the rescue effect (Fig. 1f). These findings highlight the critical role of Ser168 in embryonic axis induction.

To further explore the importance of Ser168 in Hwa, we tested various amino acid substitutions at this site. Threonine substitution (S168T) largely preserved the activity of *hwa* mRNA, while other substitutions almost completely negated the axis-inducing function of Hwa (Fig. 2a). This indicates that any substitution to non-serine/threonine amino acids at this site severely impairs Hwa activity in embryonic axis induction. We also examined the impact of Ser168 mutation on downstream signaling in HEK293T cells. As a measure of signaling activation, we extracted cytosolic β-catenin using digitonin and analyzed it by immunoblotting in cells transfected with different Hwa variants [wildtype, PPNSP motif depletion [Hwa(ΔPPNSP)], and single-amino acid substitutions [Hwa(S168A) & Hwa(S168E)]]. Neither Hwa(ΔPPNSP) nor Hwa(S168A/E) enhanced cytosolic β-catenin protein levels like the wildtype Hwa (Fig. 2b, c). Similarly, the Dual-Luciferase Reporter assay for SuperTop Flash showed that both Hwa(S168A) and Hwa(S168E) abolished the activity of Hwa in β-catenin signaling (Fig. 2d). Meanwhile, it remains unclear how a single amino acid mutation impedes downstream signal activation. Coimmunoprecipitation experiments showed Hwa(ΔPPNSP) and Hwa(S168A) had reduced interactions with human tankyrase 1 (TNKS1) compared to wild-type Hwa (Fig. 2e). Additionally, the S168A mutation prevented Hwa-mediated Axin1 protein degradation (Fig. 2f, g), although its own stability of remained largely unaffected (Supplementary Fig. 1). Collectively, these results suggest that substituting Ser168 or deleting the PPNSP motif weakens Tankyrase interaction and recruitment, diminishes Axin degradation, and disrupts cytosolic β-catenin stability, thus impeding signaling activation.

### Ser168 of Hwa is phosphorylated in HEK293T cells and zebrafish embryos
It is noteworthy that only the threonine substitution at the Ser168 site retains the Hwa activity (Fig. 2a). Serine and threonine typically act as phosphate group acceptors, undergoing phosphorylation by protein kinases for signal transduction. This suggests Hwa may function as a signal transducer, activated via phosphorylation. To identify the posttranslational modifications, Flag-tagged Hwa protein, expressed in HEK293T cells, was immunoprecipitated and enriched. Subsequently, the target band was excised from a Coomassie brilliant blue-stained SDS-PAGE gel for analysis via liquid chromatography coupled to tandem mass spectrometry (LC-MS/MS). The primary analysis revealed multiple phosphorylation sites on the Hwa protein, including Ser168.

To verify the phosphorylation state of Ser168, nonphosphorylated (VNTVPPNSPVLR) and phosphorylated (VNTVPPN(p)SPVLR) peptides were synthesized based on the primary MS data. They served as unphosphorylated and phosphorylated controls, respectively (Fig. 3a, b). Representative secondary mass spectra from Hwa protein derived from HEK293T cells and zebrafish embryos showed profiles similar to the synthetic phosphorylated peptide (Fig. 3c, d). In summary, the LC-MS/MS results confirmed that Ser168 of Hwa is phosphorylated in both mammalian and zebrafish cells, likely influencing Hwa's activity.

To quantify the phosphorylation state, an antibody, pHwa(Ser168) or simply pHwa, was developed to specifically recognize phosphorylated Ser168 of Hwa. The specificity of the antibody, purified from #11485 serum, was confirmed through phosphatase treatment and mutant Hwa proteins. This specific band (pHwa) disappeared in samples pretreated with λ-PPase but was partially restored with phosphatase inhibitor co-treatment, indicating the pHwa antibody uniquely recognizes phosphorylated Hwa (Supplementary Fig. 2a, b). Additionally, the pHwa antibody identified a specific band in only Hwa(WT) samples but not in Hwa(ΔPPNSP), Hwa(S168E) and Hwa(S168A) mutants (Supplementary Fig. 2c–f) both in zebrafish and HEK293T cells.

### Multiple kinases can phosphorylate Ser168 of Hwa in HEK293T cells and in vitro
With the specific antibody, we attempted to identify the kinase responsible for phosphorylating Hwa at Ser168. First, in silico kinase

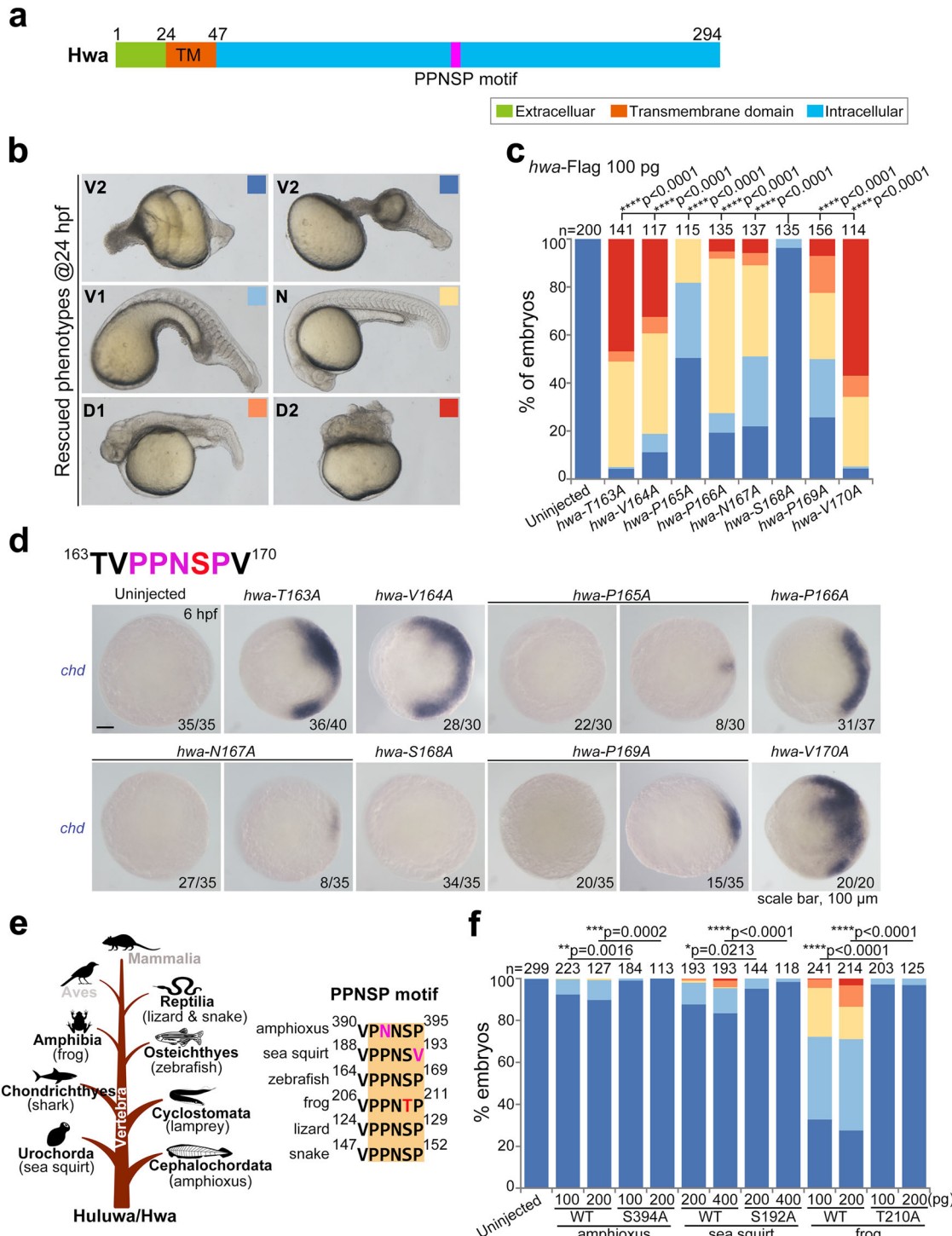

**Fig. 1 | Ser168 is indispensable for the axis-inducing activity of Hwa.**
**a** Domain and motif composition of zebrafish Hwa protein. **b** Phenotypes of
M*hwa*$^{tsuO1sm/tsuO1sm}$ embryos rescued by different mRNAs. These were grouped into
five classes from ventralized to dorsalized at 24 hpf (V2 < V1 < N < D1 < D2; V,
ventralized; N, normal; D, dorsalized), indicated by squares with different colors.
**c** The rescue efficiency of different point-mutated mRNAs of *hwa* (163-170Aa)
was plotted according to the classification criteria in (**b**), N = 2. **d** The upper panel
shows the 163-170 aa sequence of Hwa, including the conserved PPNSP motif;
the lower panel presents the expression of the dorsal marker gene *chd* in
M*hwa*$^{tsuO1sm/tsuO1sm}$ embryos rescued by different mutant mRNAs of *hwa*, injected
at the 1-cell stage and harvested at 6 hpf. All figures were imaged from the animal
view, with numbers at the right corner. Scale bar, 100 μm; n, numbers of

embryos. **e** The left panel indicates the presence of Hwa protein in the phylum
Chordata excluding two classes (Aves and Mammalia are marked in gray color);
the right panel shows the conservation of the PPNSP motif in different species.
**f** The rescue efficiency of wild type (WT) and point-mutation *hwa* mRNAs from
different species in zebrafish M*hwa*$^{tsuO1sm/tsuO1sm}$ embryos according to the classi-
fication criteria in (**b**), N = 3. 100 pg of each *hwa*-Flag mRNA (**b**–**d**) was injected
per embryo at the 1-cell stage. (**c**–**f**) A two-tailed Fisher's exact test was per-
formed to evaluate differences between treatments (all phenotypes were divided
into two groups: Unchanged [V2] and Changed [V1-D2]). N, number of biological
replicates; n, total number of embryos in each treatment; Significant differences
are indicated by ns ≥ 0.05, *p < 0.05, **p < 0.01, ***p < 0.001, and ****p < 0.0001.
Source data are provided as a Source Data file.

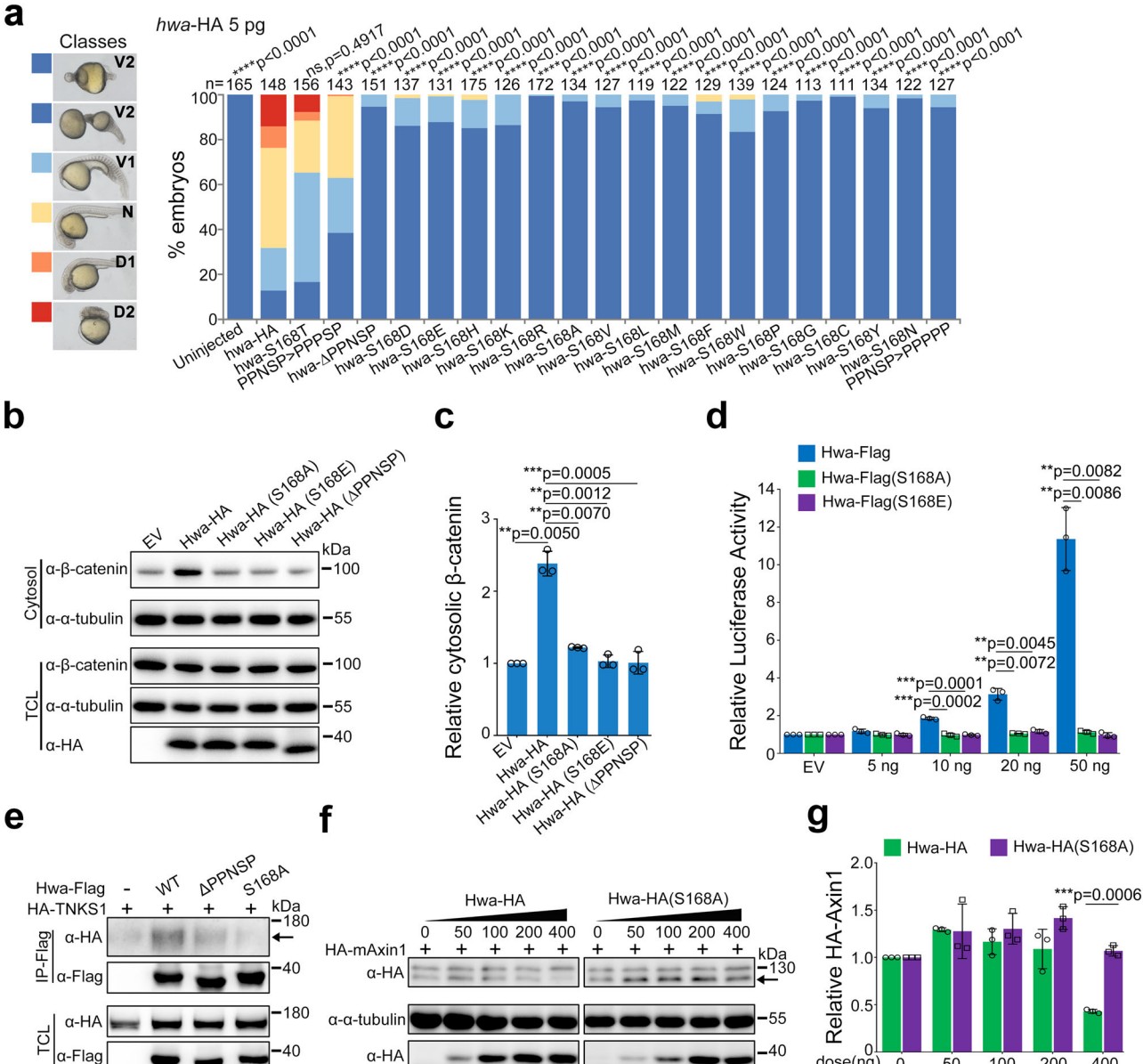

**Fig. 2 | Mutation at Ser168 of Hwa attenuates the activation of β-catenin signaling. a** The left panel indicates the classification of rescued phenotypes at 24 hpf (V2 < V1 < N < D1 < D2; V, ventralized; N, normal; D, dorsalized); the right panel shows the rescue efficiency of wild-type and single-amino acid-substituted *hwa*-HA mRNAs in M*hwa^tsu01sm/tsu01sm* embryos, with 5 pg of each *hwa*-HA mRNA injected per embryo at the 1-cell stage, *N* = 2. **b** Immunoblotting of β-catenin from the cytosol (active form) and total cell lysate (TCL) of HEK293T cells overexpressing wild-type or mutant Hwa-HA protein. PPNSP motif deletion or Ser168 mutation nearly abolished the activation of the β-catenin signal by Hwa. **c** Quantifications of relative cytosolic β-catenin levels in HEK293T cells treated as in (**b**), *N* = 3. **d** SuperTop Flash was applied to check the β-catenin signal-inducing activity of wild-type and Ser168 mutant Hwa (S168A and S168E), *N* = 3. **e** Coimmunoprecipitation of HA-TNKS1 with wild-type and mutant Hwa-Flag proteins. Protein lysates were immunoprecipitated with anti-Flag antibodies; the arrow indicates the HA-TNKS1 protein, *N* = 3. **f** Immunoblotting of HA-Axin1 in cells co-transfected with different doses of wild-type or S168A Hwa-HA plasmids. The arrow indicates the HA-mAxin1 protein. **g** Quantifications of relative HA-Axin1 protein levels in HEK293T cells treated as in (**b**), *N* = 3. α-tubulin (**b**) or total HA-TNKS1 (**f**) was used as references, and relative protein levels are indicated in (**c**) and (**g**). **a** A two-tailed Fisher's exact test was performed to evaluate differences between treatments (all phenotypes were divided into two groups: Unchanged [V2] and Changed [V1-D2]). (**c**–**g**) A two-tailed unpaired *t*-test was performed and data were presented as mean ± SD. N, number of biological replicates; n, total number of embryos in each treatment; Significant differences are indicated by ns ≥ 0.05, *p < 0.05, **p < 0.01, ***p < 0.001, and ****p < 0.0001. Source data are provided as a Source Data file.

prediction on the website (http://kinasephos.mbc.nctu.edu.tw/predict.php) revealed that the PPNSP motif includes the consensus sequence (serine-proline (SP)) of substrates of CDK (Cyclin-dependent kinase). This finding suggests that CDK may be involved in Ser168 phosphorylation (Supplementary Fig. 3a). To test this hypothesis, we applied two inhibitors (AT7519 and AZD5348) to block endogenous CDKs in HEK293T cells. The pHwa levels decreased significantly in a dose-dependent manner, while the total Hwa protein level remained largely unchanged (Supplementary Fig. 3b). This result indicates a potential role of CDK in Hwa phosphorylation.

Hwa is a transmembrane protein harboring an intracellular PPNSP motif, which is similar to the PPPS/TP motifs seen in LRP6. The first PPPSP motif (Ser1490) of Lrp6 is phosphorylated by Cdk14 in *Xenopus*. Furthermore, Cdk14 and Cdk16 belong to the same subfamily, and they were previously reported to be activated by membrane-targeted cyclins (Cyclin Y [Ccny]) or (Cyclin Y like 1 [Ccnyl1]), respectively[23–25]. Based on

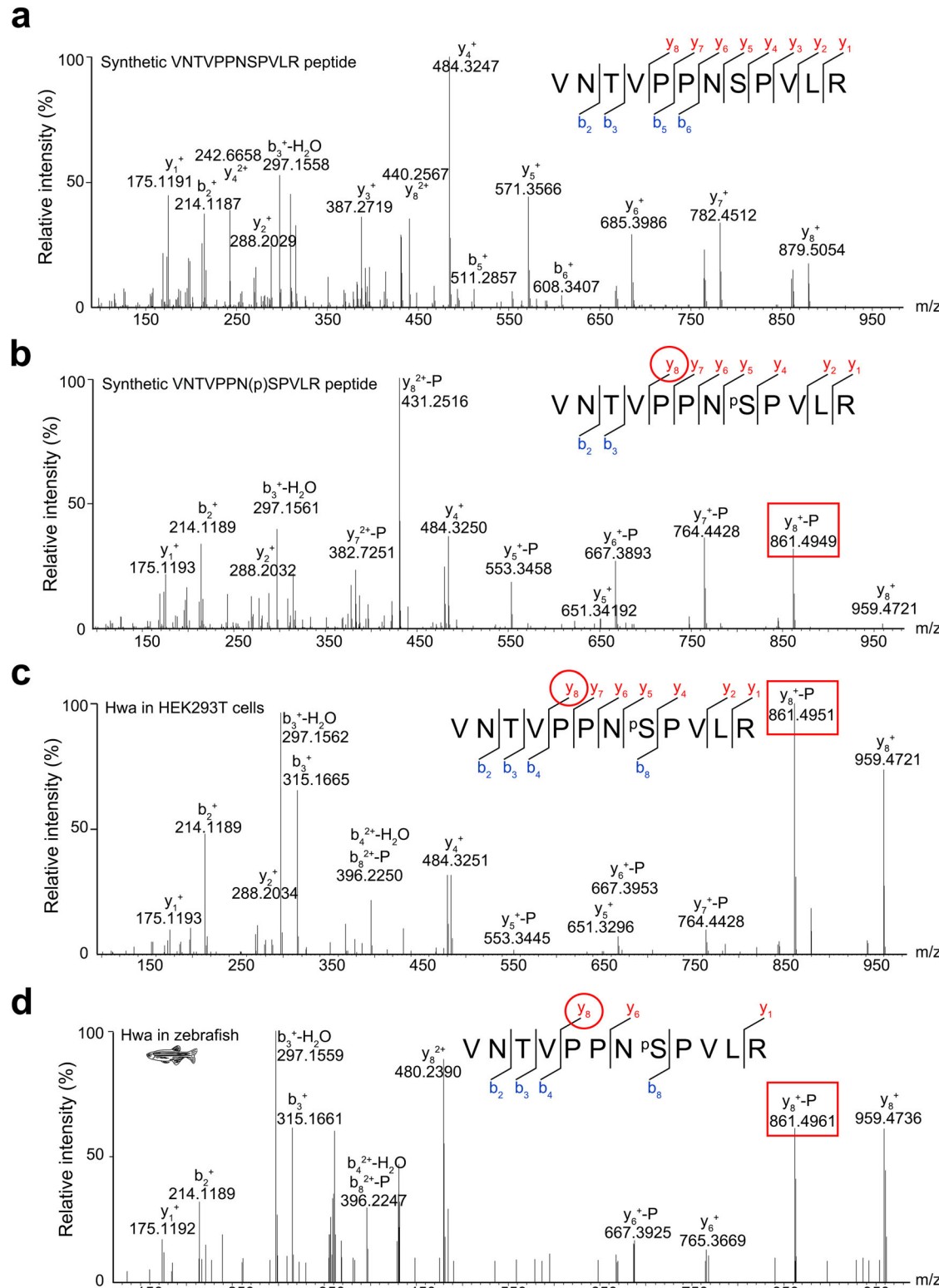

**Fig. 3 | The phosphorylation of Hwa protein at Ser168 was validated by LC-MS/ MS.** Synthetic nonphosphorylated (**a**) and phosphorylated (**b**) peptides were used as references for the nonphosphorylated and phosphorylated states, respectively. Flag-tagged Hwa proteins purified from HEK293T cells (**c**) and zebrafish embryos (**d**) were used for phosphorylation identification, and the MS spectra of fragments containing VNTVPPNSPVLR were compared with that of phosphorylated peptide (**b**) to ensure phosphorylation at the Ser168 site. The labels "y" and "b" designate the C- and N-terminal peptide fragment ions, respectively, which were produced by collision-induced fragmentation at the peptide bond. The subscripted number (e.g., y8, b2,) represents the number of C- or N-terminal residues in the peptide fragment. The labels "-$H_2O$" and "-P" designate ions with water ($H_2O$) and phosphoric acid ($H_3PO_4$) loss, respectively. The red circle indicates the specific collision-induced fragmentation to identify the phosphorylation state and the red rectangle indicates the corresponding peak in the MS profile. Source data are provided as a Source Data file.

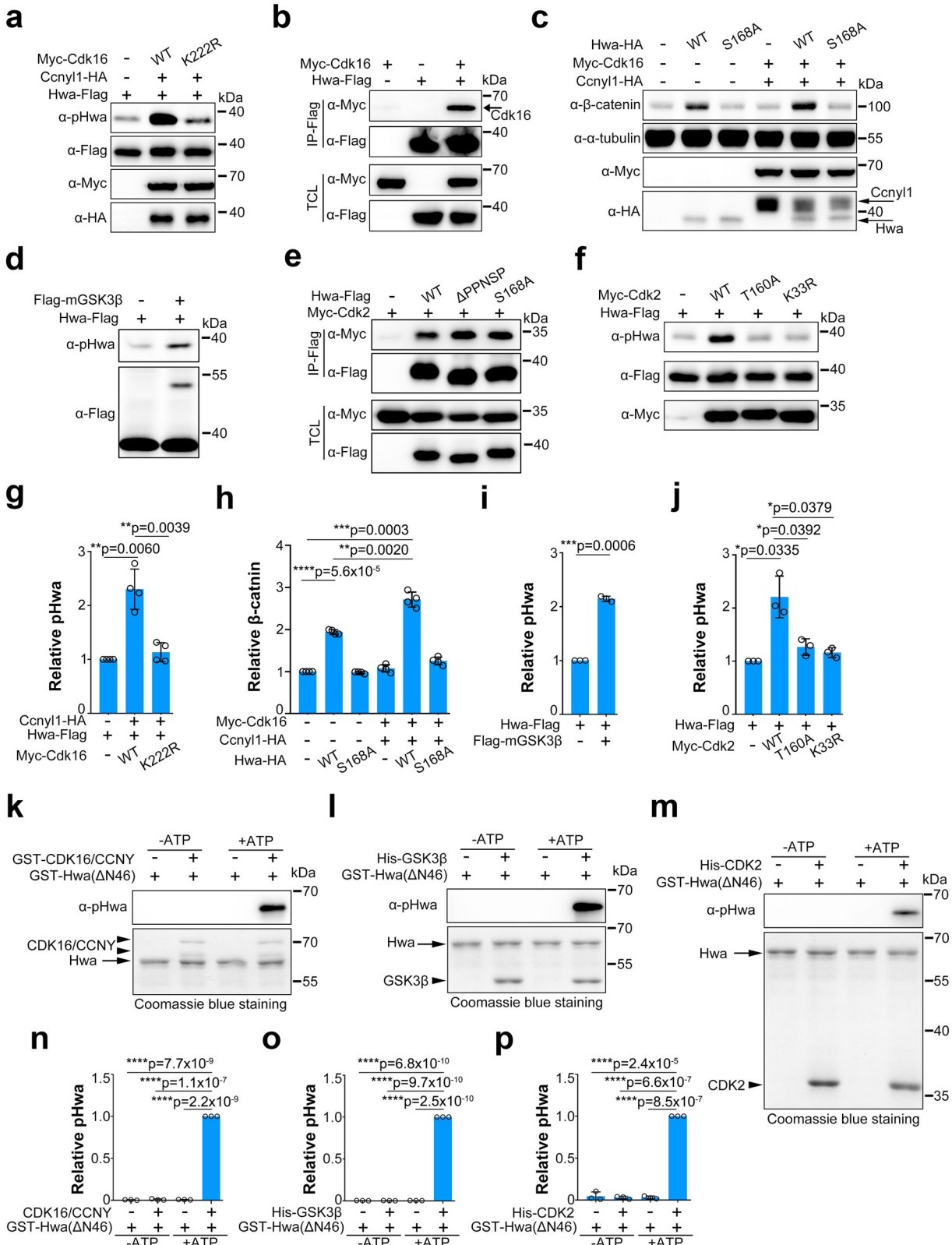

this, we firstly evaluated whether Cdk14 and/or Cdk16 are responsible for Ser168 phosphorylation. Interestingly, Cdk16 but not Cdk14 increased Ser168 phosphorylation in Hwa when coexpressed with Ccny or Ccnyl1 (Supplementary Fig. 4). In addition, this enhancement of pHwa was Ccny/Ccnyl1 dose-dependent (Supplementary Fig. 5a-b).

However, the kinase-dead form of Cdk16(K222R) (similar to the K33R mutation in CDK2[26]) failed to phosphorylate Hwa (Fig. 4a), and

AZD5438 inhibited the phosphorylation by Cdk16/Ccnyl1, which further confirmed the dependence of kinase activity for phosphorylation (Supplementary Fig. 5c, d). Furthermore, the coimmunoprecipitation results showed that Cdk16 interacted with Hwa (Fig. 4b), and this interaction decreased in the presence of Ccnyl1-HA, but increased when used the kinase-dead form of Cdk16(K222R) (Supplementary Fig. 5e, f). A previous study suggested that Ccny/Ccnyl1 interacts with

**Fig. 4 | Ser168 of Hwa can be phosphorylated by multiple kinases in HEK293T cells and in vitro. a** Immunoblotting of pHwa from HEK293T cells cotransfected with Ccnyl1-HA and wild-type and the kinase-dead form of Myc-Cdk16(K222R). **b** Coimmunoprecipitation of Hwa-Flag with Myc-Cdk16, N = 3. **c** Immunoblotting of cytosolic/active β-catenin in HEK293T cells transfected with Hwa-Flag (WT, S168A) alone or with Myc-Cdk16/Ccnyl1-HA. **d** Immunoblotting of pHwa from HEK293T cells transfected with Hwa-Flag and Flag-mGSK3β (membrane-tagged GSK3β). **e** Coimmunoprecipitation of Myc-Cdk2 with different forms of Hwa-Flag proteins (WT, S168A and ΔPPNSP), N = 3. **f** Immunoblotting of pHwa from HEK293T cells transfected with wild-type and the kinase-dead form of Myc-Cdk2(T160A, K33R). **g** Quantifications of relative pHwa levels in HEK293T cells treated as in (**a**), N = 4. **h** Quantifications of relative cytosolic/active β-catenin levels in HEK293T cells treated as in (**c**), N = 4. **i** Quantifications of relative pHwa levels in HEK293T cells treated as in (**d**), N = 3. **j** Quantifications of relative pHwa levels in

HEK293T cells treated as in (**f**), N = 3. **k** In vitro phosphorylation of purified Hwa by recombind CDK16/CCNY proteins in the absence or presence of ATP. **l** In vitro phosphorylation of purified Hwa by recombinant His-GSK3β in the absence or presence of ATP. **m** In vitro phosphorylation of purified Hwa by recombinant His-CDK2 proteins in the absence or presence of ATP. **n**–**p** Quantifications of relative pHwa levels in in vitro phosphorylation experiments treated with different kinase proteins as shown in (**k**–**m**), N = 3. The arrow and arrowhead indicate Hwa and kinase proteins, respectively (**k**–**m**). Total Hwa (**a**, **d**, **f** and **k**–**m**) and α-tubulin (**c**) proteins were used as references for quantification in immunoblotting. (**g**–**j** and **n**–**p**) A two-tailed unpaired t-test was performed and data were presented as mean ± SD. N, number of biological replicates; Significant differences are indicated by ns ≥ 0.05, *$p < 0.05$, **$p < 0.01$, ***$p < 0.001$, and ****$p < 0.0001$. Source data are provided as a Source Data file.

and activates Cdk16[25,27,28]. Our work suggested that Ccnyl1 might activate Cdk16 to facilitate Hwa binding, regardless of the catalytic activity of Cdk16. Wild-type Cdk16 can be activated by Ccny/Ccnyl1 resulting in further phosphorylation of Hwa at Ser168, and the phosphorylation of Ser168 may decrease the binding affinity between Cdk16 and pHwa. In the case of Cdk16 (K222R), phosphorylation results in increased binding to Hwa in the presence of Ccnyl1, although Hwa is not able to be phosphorylated and released.

Consistent with the reported function of Ccny/Ccnyl1[23–25], zebrafish Ccny protein recruited Cdk16 to the plasma membrane in both zebrafish embryos and HEK293T cells (Supplementary Fig. 6). Functionally, Cdk16/Ccnyl1 overexpression elevated the activity of Hwa in downstream signaling, as indicated by the cytosolic β-catenin levels in HEK293T cells (Fig. 4c and h). In summary, Cdk16 kinase phosphorylates Hwa at Ser168 and activates downstream signaling, which is regulated by Ccny/Ccnyl1.

Regarding GSK3β, a kinase known to phosphorylate Ser1490 in the PPPSP motif of the Lrp6 coreceptor[18], we then investigated whether it could also phosphorylate Ser168. Although GSK3β controls both the on and off states of β-catenin signaling, with activation by the membrane-associated form, and inhibition by cytosolic GSK3[18], we found both the membrane-associated GSK3β (mGSK3β) and cytosolic GSK3β enhanced the Ser168 phosphorylation in Hwa (Fig. 4d, i and Supplementary Fig. 7a, b). This result led us to consider the contribution of other kinases. CDKs, part of the CMGC family, consist of various subfamilies in mammals (Supplementary Fig. 8a, b). Nearly twenty cdk genes are expressed in zebrafish embryos (Supplementary Fig. 8c). We examined several widely expressed CDKs (Cdk1, Cdk2, Cdk4, and Cdk6), and found that Cdk2 interacted with Hwa (Fig. 4e and Supplementary Fig. 7g) and enhanced the Ser168 phosphorylation (Supplementary Fig. 7c–f). In contrast, kinase-dead forms (K33R and T160A) lost this effect (Fig. 4f). Similar to Cdk16, Cdk2 enhanced Hwa-mediated β-catenin signaling in HEK293T cells (Supplementary Fig. 7h, i).

We then conducted in vitro phosphorylation assays to check whether the phosphorylation of Ser168 by Cdk2/Cdk16 and GSK3β is direct or indirect. Purified human CDK16/CCNY protein mixture and GSK3β protein from Sf9 cells phosphorylated Glutathione S-transferase (GST)-Hwa (ΔN46) derived from E. coli in the presence of ATP (Fig. 4k, l, n, and o), indicating that CDK16/CCNY and GSK3β phosphorylate Hwa at Ser168 directly. Consistently, purified human CDK2 protein from E. coli also phosphorylated GST-Hwa(ΔN46) derived from E. coli directly in the presence of ATP (Fig. 4m, p). In summary, multiple kinases contribute to Ser168 phosphorylation in the Hwa protein.

**Ser168 of Hwa is phosphorylated in zebrafish and is responsible for axis induction**

As the endogenous expression level of Hwa is quite low, we expressed both the wild-type and S168A mutant proteins in embryos via mRNA

microinjection to investigate whether the Hwa protein is phosphorylated in zebrafish. The immunoblotting results showed that both types of mRNA produced approximately equal amounts of Hwa protein, while pHwa was only detected in wild-type mRNA injected embryos (Fig. 5a, b). Furthermore, the pHwa level decreased significantly when the embryos were treated with AZD5348 (an inhibitor targeting Cdks and Gsk3α/β) (Fig. 5c, d). To assess the effect of Hwa phosphorylation on embryonic axis induction, we performed overexpression and rescue experiments in zebrafish embryos. cdk16 mRNA, when co-injected with ccnyl1 mRNA in WT embryos, resulted in dorsalized embryos, with some even developing double head/axis (DH) (Fig. 5e). Additionally, co-injecting cdk16/ccnyl1 mRNA with hwa mRNA in Mhwa$^{tsuO1sm/tsuO1sm}$ mutant embryos showed higher rescue efficiency than hwa mRNA alone (Fig. 5f). This was further evidenced by the expression of the dorsal organizer marker genes boz and chd at 4 hpf (Fig. 5g). However, injecting cdk16/ccnyl1 mRNA without hwa mRNA had little or no effect on axis formation, aligning with results in HEK293T cells (Supplementary Fig. 5a and 9a). Similarly, co-injecting hwa mRNA with mGSK3β and cdk2 mRNA in Mhwa$^{tsuO1sm/tsuO1sm}$ mutant embryos also improved rescue efficiency and expression of organizer genes compared to hwa mRNA alone (Fig. 5h–j). These findings demonstrate that overexpressing kinases responsible for Ser168 phosphorylation enhances the axis-inducing activity of Hwa.

**Attenuating phosphorylation at Ser168 blocks the axis-inducing activity of Hwa**

Elevating phosphorylation at Ser168 enhances Hwa's function in β-catenin signaling and axis induction. We next examined what would occur if Hwa's phosphorylation is disrupted. Initially, we tried to block the phosphorylation of Hwa in embryos with inhibitor AZD5438, but both WT and Mhwa$^{tsuO1sm/tsuO1sm}$ mutant embryos exhibited dorsalized phenotypes after treatment. This could have resulted from the known multiple effects of the inhibitor, namely, activating maternal β-catenin signaling via blocking Gsk3β and inhibiting zygotic Wnt/β-catenin and BMP signaling pathways[29–31], which are necessary for ventral fate. Therefore, a more specific strategy is needed to block the phosphorylation process. It was reported that the D145N mutation in human CDK2 functions as a dominant negative (DN) form and blocks the progression of the cell cycle[32]. Human CDK16 and zebrafish Cdk16 are highly conserved at this site compared to CDK2/Cdk2 proteins (Fig. 6a). Therefore, D-to-N substitution in the corresponding position may function as a dominant negative form of Cdk16. Indeed, when coexpressed in HEK293T cells, Cdk16$^{DN}$ attenuated the phosphorylation of Hwa at Ser168 by wild-type Cdk16 in a dose-dependent manner (Fig. 6b, c).

When coinjected with hwa mRNA in Mhwa$^{tsuO1sm/tsuO1sm}$ mutant embryos, cdk16$^{DN}$ mRNA decreased the expression of organizer/dorsal markers (boz at 4 hpf and chd at 6 hpf) (Fig. 6d, e). Consistently, cdk16$^{DN}$ mRNA alone or with ccnyl1 mRNA attenuated the axis-inducing activity of Hwa (Fig. 6f), especially when hwa mRNA was injected at a

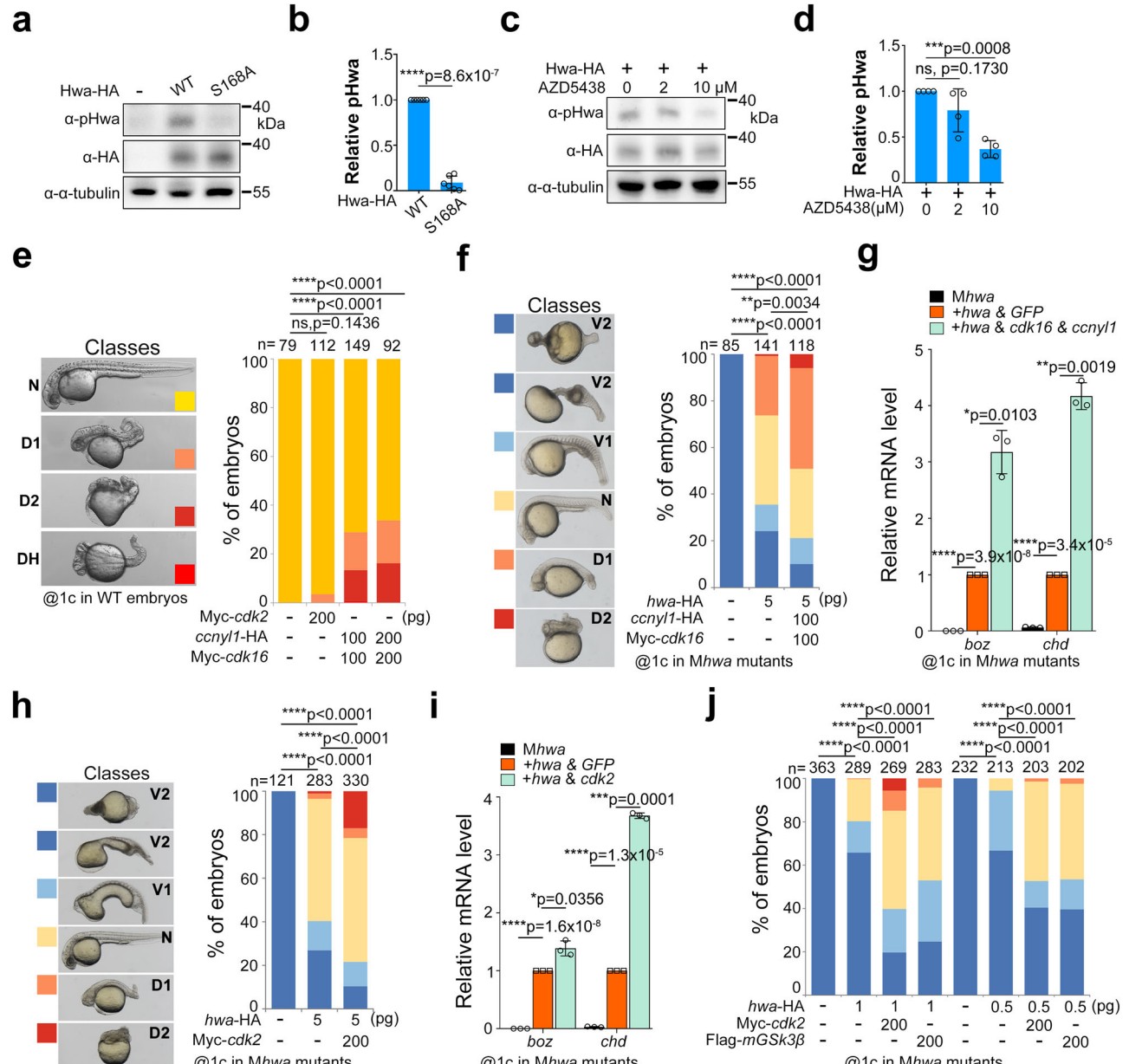

**Fig. 5 | Ser168 of Hwa is phosphorylated in zebrafish and is responsible for axis-inducing activity. a** Immunoblotting of pHwa in zebrafish embryos injected with 200 pg WT or S168A mutant of *hwa*-HA mRNA at the 1-cell stage. **b** Quantifications of relative pHwa levels in embryos treated as in (**a**), $N = 6$. **c** Immunoblotting of pHwa in zebrafish injected with 200 pg *hwa*-HA mRNA at the 1-cell stage, followed by treatment with AZD5438 from 2–4 hpf. (**d**) Quantifications of relative pHwa levels in embryos treated as in (**c**), $N = 4$. **e** Overexpression of Myc-*cdk16* and *ccnyl1*-HA in wild-type embryos resulted in dorsalized phenotypes (D1, D2), some with double head/axis (DH), $N = 2$. **f** Rescue efficiency of *hwa*-HA mRNA alone or together with Myc-*cdk16* & *ccnyl1*-HA injected at the 1-cell stage in M*hwa*^tsu01sm/tsu01sm^ embryos, $N = 2$. **g** Expression levels of the organizer-specific genes (*boz* and *chd*) in embryos rescued by different mRNA combinations of *hwa*-HA and Myc-*cdk16/ccnyl1*-HA as in (**f**), $N = 3$. **h** Rescue efficiency of 5 pg *hwa*-HA mRNA alone or together with Myc-*cdk2* injected at the 1-cell stage in M*hwa*^tsu01sm/tsu01sm^ embryos, $N = 3$. **i** Expression levels of *boz* and *chd* in embryos rescued by different mRNA combinations of *hwa*-HA and Myc-*cdk2* as in (**h**), $N = 3$. **j** Rescue efficiency of lower dose (1.0 or 0.5 pg) of *hwa*-HA mRNA alone or together with Flag-*mGSK3β* and Myc-*cdk2* injected at the 1-cell stage in M*hwa*^tsu01sm/tsu01sm^ embryos, $N = 3$. phenotypes were grouped as in (**h**). **b**–**i** A two-tailed unpaired *t*-test was performed; (**e**–**j**) A two-tailed Fisher's exact test was performed (all phenotypes were divided into two groups: Unchanged and Changed). N, number of biological replicates; n, total number of embryos in each treatment; Significant differences are indicated by ns ≥ 0.05, *$p < 0.05$, **$p < 0.01$, ***$p < 0.001$, and ****$p < 0.0001$. Source data are provided as a Source Data file.

lower dose (Supplementary Fig. 9a). When injected into WT zebrafish embryos, *cdk2*^DN^ mRNA caused extensive apoptosis but had little effect on axis induction (Supplementary Fig. 10a) due to cell cycle inhibition[32]. *cdk16*^DN^ mRNA injection did not affect cell survival but resulted in some ventralized embryos, albeit with a relatively low proportion (Supplementary Fig. 9b and 10b). These results demonstrated that the D277N substitution of zebrafish Cdk16 functions as a

dominant negative form and competitive inhibitor of phosphorylation and activation of the Hwa protein. Furthermore, the pHwa antibody was functional in vivo, which attenuated dorsal axis formation and led to a ventralized phenotype in 20–30% of embryos when injected at the 1-cell stage (Fig. 6g). Injected antibodies may bind to phosphorylated Hwa, preventing the interaction of other molecules and block downstream signal transduction.

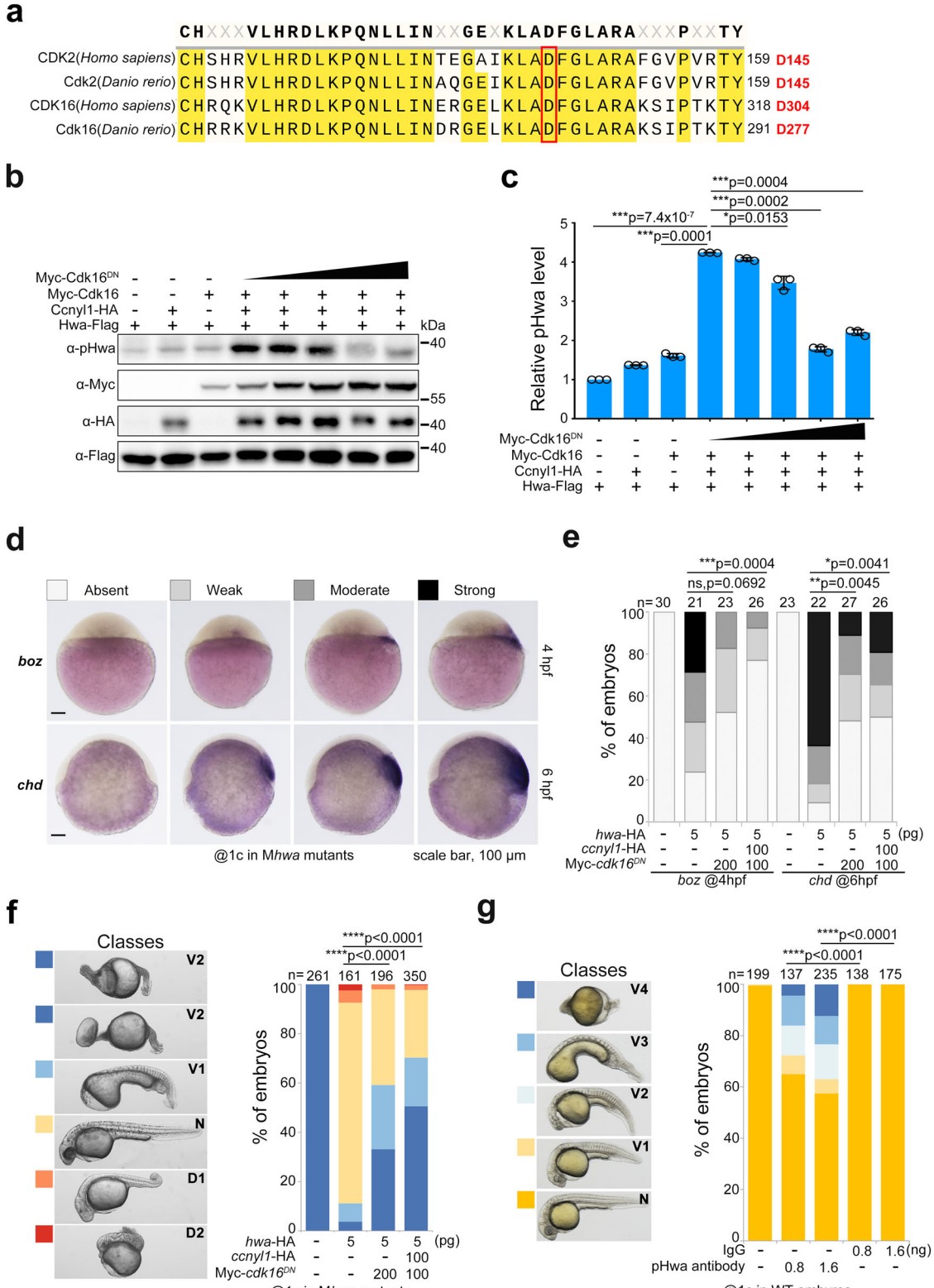

## S168A mutation of Hwa causes loss of the embryonic axis in zebrafish

To elucidate the endogenous role of Ser168 in zebrafish embryos, a mutant line with a serine-to-alanine (S168A) substitution was created using CRISPR/Cas9 technology (Fig. 7a). To assess the S168A mutation's functional effect, the S168A/S168A homozygous females were raised and genotyped (Fig. 7b). Embryos from heterozygous females (M$hwa^{+/S168A}$) developed normally, but embryos from homozygous females (M$hwa^{S168A/S168A}$) exhibited phenotypes identical to M$hwa^{tsu01sm/tsu01sm}$, divided into two types: Class I with an onion shape (69/157) and Class II with a calabash shape (88/157). Both classes are radially symmetric and lack a body axis (Fig. 7c). Consistent with the phenotype, nearly no expression of dorsal/organizer marker genes (*boz* at 4 hpf and *chd/gsc* at 6 hpf) while increased ventral fate genes (*vox* at 4 hpf and *bmp2b/vent* at 6 hpf) were detected in M$hwa^{168A/S168A}$ embryos (Fig. 7d), which was further confirmed by WISH (Fig. 7e). The

**Fig. 6 | Attenuating phosphorylation of Ser168 by Cdk16^(DN) or pHwa antibody disrupts the axis-inducing activity of Hwa. a** Residue conservation analysis of human CDK2/16 and zebrafish Cdk2/16 proteins. The red box indicates the conserved, functionally critical aspartic acid (D). **b** Immunoblotting of pHwa in HEK293T cells transfected with Hwa-Falg, wild-type Cdk16, and different doses of dominant negative Cdk6 (Myc-Cdk16^(DN)). **c** Quantifications of relative pHwa levels in HEK293T cells treated as in (**b**), N = 3.Total Hwa protein was used as an internal control. **d** The WISH results of *boz* and *chd* in *cdk16^(DN)* mRNA injected embryos at 4 hpf and 6 hpf, respectively. **e** The statistical results of embryos treated as in (**d**). **f** Effect of coexpression of *cdk16^(DN)* and *ccnyl1*-HA mRNA with *hwa* mRNA in

M*hwa^(tsu01sm)* embryos, N = 3. **g** Phenotypes of wild-type embryos injected with pHwa antibody or IgG at different doses, N = 4. Scale bars, 100 μm; V, ventralized; N, normal; D, dorsalized. Total Hwa proteins were used as references for quantification in (**c**). **c** A two-tailed unpaired *t*-test was performed and data were presented as mean ± SD. **e**–**g** A two-tailed Fisher's exact test was performed to evaluate differences between treatments (all phenotypes were divided into two groups: Unchanged and Changed). N, number of biological replicates; n, total number of embryos in each treatment; Significant differences are indicated by ns ≥ 0.05, *$p < 0.05$, **$p < 0.01$, ***$p < 0.001$, and ****$p < 0.0001$. Source data are provided as a Source Data file.

above results suggest the S168A mutation resulted in loss of body axis in zebrafish embryos.

What's more, in genetic complementation experiment, another *hwa^(S168A/tsu01sm)* compound heterozygous line further validate the effect of S168A mutation. The *hwa^(tsu01sm)* allele was previously identified as a loss-of-function variant, exhibiting minimal to no *hwa* transcripts. Sanger sequencing revealed only *hwa^(S168A)* mRNA (GCA) can be detected in embryos from M*hwa^(S168A/tsu01sm)* females with a heterozygous (TCA/GCA) genotype (Supplementary Fig. 11a). Compared to wild-type (WT) embryos, *hwa* expression level in M*hwa^(S168A/tsu01sm)* and M*hwa^(+/tsu01sm)* embryos was lower but similar (Supplementary Fig. 11b, c). Sequencing and quantification showed that the *hwa^(S168A)* allele is transcribed at levels comparable to the wild-type allele, whereas the *hwa^(tsu01sm)* allele has minimal transcript. Similar to M*hwa^(168A/S168A)* and M*hwa^(tsu01sm/tsu01sm)* homozygous embryos, M*hwa^(S168A/tsu01sm)* embryos also presented as radially symmetric and body-axis-lacking phenotypes (Supplementary Fig. 11d). WISH and RT-qPCR of dorsal and ventral marker genes further verified the loss of organizer/dorsal fate in M*hwa^(S168A/tsu01sm)* embryos (Supplementary Fig. 11e, f). These findings indicate that the S168A substitution acts as a loss-of-function allele of Hwa, underscoring the crucial role of Ser168 in Hwa's axis-inducing activity.

## Discussion

Dorsal organizer formation and embryonic axis induction are crucial in establishing cell fate in vertebrates. Previously, the maternal-effect factor Hwa was identified as a determinant of the dorsal organizer by activating β-catenin signaling in zebrafish and *Xenopus*[15]. This work characterized the functionally critical Ser168 of Hwa for signal activation. The deletion or substitution of Ser168 significantly disrupted axis induction and β-catenin signaling. Through LC-MS/MS characterization and immunoblotting with a phosphorylation-specific antibody, Ser168 was found to be phosphorylated in vivo. Additionally, this site's phosphorylation level positively correlated with the β-catenin signal. Furthermore, at least Cdk16, Cdk2, and GSK3β promoted Ser168 phosphorylation in HEK293T cells. An in vitro assay with purified proteins indicated that corresponding kinases phosphorylate Hwa at Ser168 directly. In zebrafish embryos, pHwa was detected in WT but not S168A mutant mRNA-injected embryos and decreased dramatically when the injected embryos were treated with AZD5438 to inhibit endogenous Cdks and Gsk3α/β. *cdk2, mGSK3β*, and *cdk16+ccnyl1* mRNA injection enhanced Hwa's axis-inducing activity when co-injected with *hwa* mRNA in M*hwa^(tsu01sm/tsu01sm)* mutant embryos. Conversely, the dominant-negative form of Cdk16 (Cdk16^(DN)) and pHwa antibody injection reduced Hwa's axis-inducing function. Overall, we found that Ser168 functions as a phosphorylation switch of the Hwa protein for embryonic axis induction in zebrafish, modulated by multiple kinases (Fig. 7f).

It has been reported that phosphorylating the PPP(S/T)P motifs of Lrp5/6 is essential for activating canonical Wnt/β-catenin signaling[17–22] either in a Wnt ligands-dependent or -independent fashion by different kinases, e.g., membrane-targeted GSK3β[18], Grk5/6[33], Pka[34] and Pftk1/Cdk14[24,35]. Cdk14, in cooperation with Ccny or Ccnyl1, cyclins tethered to the plasma membrane via N-myristoylation[23], phosphorylates Ser1490 of Lrp6. This phosphorylation is constitutive but Wnt-

independent, peaks at the G2/M phase of the cell cycle and results in enhanced β-catenin signaling[29]. This enhanced Wnt/β-catenin signal in the G2/M phase is indispensable for dorsal-ventral patterning in *Xenopus*; knockdown/depletion of maternal *ccny* with Morpholino reduced maternal Lrp6 phosphorylation at Ser1490 and inhibited expression of dorsal marker genes, *siamois* and *Xnr3*[24]. Previously, we demonstrated that Hwa works independently of Wnt ligands and receptors. Also, Hwa could rescue the phenotype of Lrp6 depletion in *Xenopus*. It is possible that Ccny/Cdk14 regulates the Lrp6 phosphorylation in oocytes to regulate proper translation, translocation, or deposition of the dorsal determinant Hwa. Hwa harbors a basic arginine at the +4 position following serine-proline (SP), which is a unique preference sequence of Cdk16 compared to other Cdks. Therefore, Cdk16 showed stronger activity than Cdk14 in the phosphorylation of Ser168 of Hwa (Supplementary Fig. 4). When the PPNSP motif in Hwa was replaced by the PPPSP motif in Lrp6, it impaired the axis-inducing activity of Hwa to some extent (Fig. 2a). This indicates that although Hwa and Lrp5/6 harbor similar motifs (PPNSPVLR & PPP(S/T)PxS/T) in intracellular domains, they could have different preferred kinases. It is not clear whether this helps distinguish maternal and zygotic β-catenin signaling (zygotic signaling is dependent on Wnt ligand stimulation and Lrp5/6 activation). Therefore, the detailed mechanism needs further study.

Substituting serine or threonine with aspartic acid (D) or glutamic acid (E), which introduces a negative charge and alters the conformation, is a common strategy to mimic phosphorylation states[36–38]. In Hwa's Ser168 case, D/E substitution negated the axis-inducing activity and function of Hwa, similar to Alanine substitution. Several possible explanations are following: the D/E substitution at Ser168 only partially mimics the phospho-group; the substitution affects Hwa protein's folding or conformation; Ser168 and surrounding serine/threonine are phosphorylated in a specific order, with the substitution disrupting this and affecting adjacent phosphorylation sites; and the phosphorylation at Ser168 is an intermediate dynamic state in signal transduction. Nonetheless, Ser168 is vital for the proper functioning of Hwa in axis induction.

Concerning CDK kinases, Cdk1, Cdk2, Cdk4, and Cdk6 are usually considered to be cell cycle-related. In contrast, Cdk7, Cdk8, Cdk9, Cdk11, and Cdk20 are grouped in transcriptional subfamilies. They modulate mRNA transcription via phosphorylation of the CTD domain of RNAPII[39–41]. Several Cdk kinases contain additional motifs in addition to the kinase domain for functional relevance: Cdk12/Cdk13(PITAIRE)[42,43], Cdk14/Cdk15(PFTAIRE)[23,44], and Cdk16/Cdk17/Cdk18 (PCTAIRE)[45–47]. In addition to their role in the cell cycle, Cdks coordinate with stemness[48], cell polarity[49,50], and cell fate specification/determination[51–54]. Cdk16 (also known as PCTAIRE-1) was previously reported to participate in secretory cargo transport[55], autophagy[28], spermatogenesis[28,56], and tumor proliferation[57]. In this work, we uncovered a role for Cdk16 and Cdk2 in modulating Hwa/β-catenin signaling. This role is essential for dorsal fate determination and body axis formation, highlighting the crosstalk between cell cycle-related factors and axis-inducing signaling. This is consistent with Mary C. Mullins's study, which found that cell division at the cleavage stage is linked to dorsal fate determination and axis formation in zebrafish embryos[58].

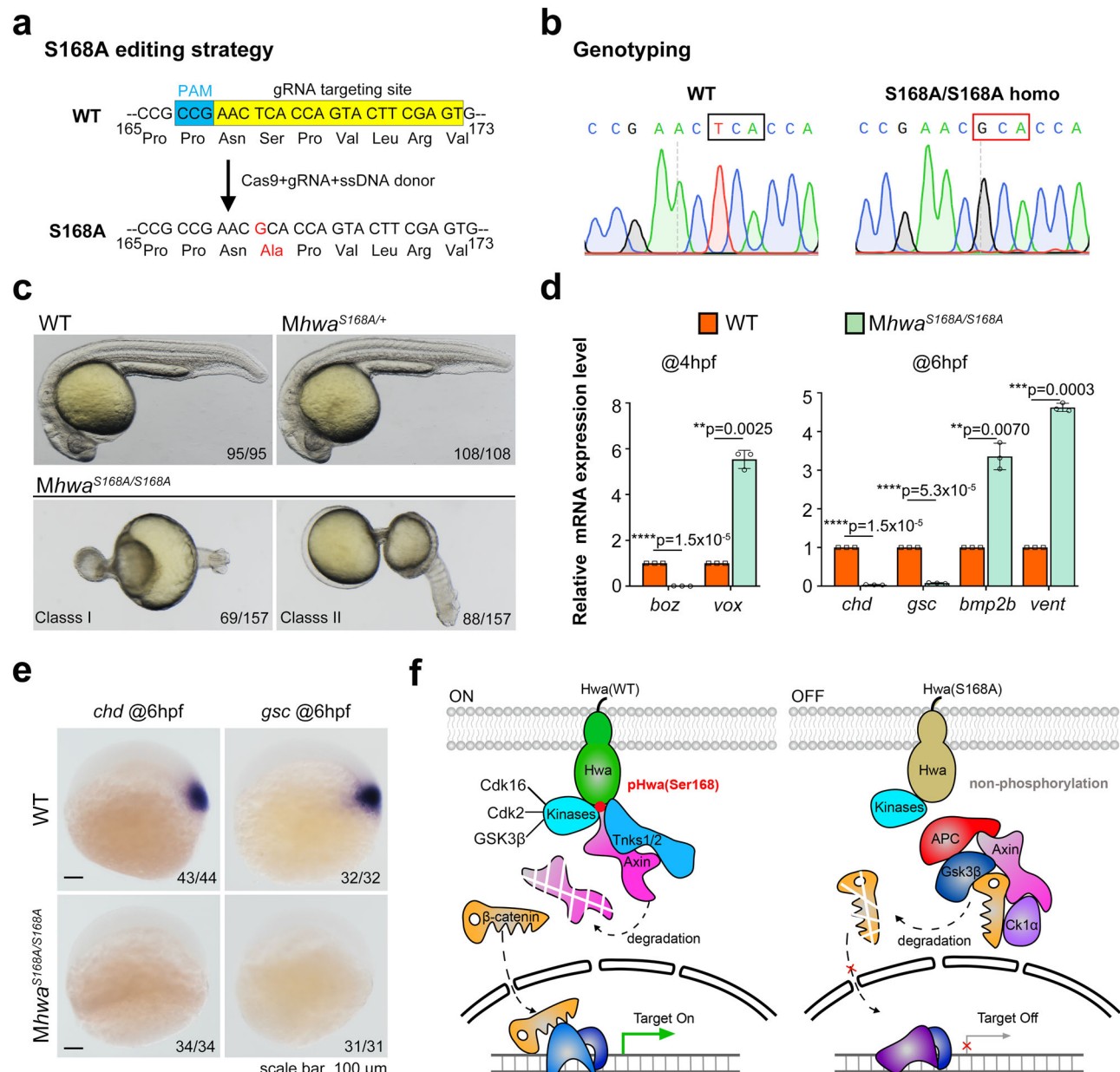

**Fig. 7 | The endogenous function of the S168A mutation in zebrafish embryos.**
**a** The CRISPR/Cas9-based editing strategy was applied to construct the S168A mutation in zebrafish embryos. The gRNA targeting site and PAM sequence are highlighted in yellow and blue, respectively; Cas9 mRNA, gRNA, and single-strand donor DNA were coinjected at the 1-cell stage. **b** Sanger sequencing of WT and M*hwa*^S168A/S168A^ females confirmed the successful construction of the *hwa*^S168A^ homozygous mutant line. **c** Phenotypes of M*hwa*^S168A/S168A^ embryos at 24 hpf could be divided into two classes: both lacking an embryonic axis, and M*hwa*^S168A/+^ embryos were all normal. **d** Marker genes were quantified by RT-qPCR at 4 hpf and 6 hpf for WT and M*hwa*^S168A/S168A^ embryos, *N* = 3. **e** Loss of expression of dorsal/organizer markers (*chd* and *gsc* at 6 hpf) was detected by WISH in M*hwa*^S168A/S168A^

embryos. **f** The working model of the Hwa receptor. In wild-type embryos, Hwa protein is phosphorylated at Ser168 by multiple kinases and activated, enhancing Tnks1/2-mediated degradation of the Axin protein, increasing stability of β-catenin in the cytosol. β-catenin then translocates into the nucleus and turns on downstream target genes. However, in M*hwa*^S168A/S186A^ mutant embryos or M*hwa*^tsu01sm/tsu01sm^ embryos supplied with Hwa(S168A) mutant protein, the point mutation switches off the activity of the Hwa receptor. **d** A two-tailed unpaired *t*-test was performed and data were presented as mean ± SD. N, number of biological replicates; Significant differences are indicated by ns ≥ 0.05, *p < 0.05, **p < 0.01, ***p < 0.001, and ****p < 0.0001, with individual *p*-values illustrated. Source data are provided as a Source Data file.

It is not clear if, other than Cdk16, Cdk2, and GSK3β, there are any other kinases involved in Hwa protein activation via Ser168 phosphorylation. Within the PCTAIRE subfamily, *cdk18* is not expressed at the early stage, but *cdk17* is both maternally and zygotically expressed, as is *cdk16* in zebrafish embryos. We attempted to establish *cdk16* and *cdk17* mutants to explore their function in zebrafish embryonic axis induction. However, while maternal-zygotic *cdk16* (MZ*cdk16*) mutant embryos develop normally, Z*cdk17* and MZ*cdk16; Zcdk17* females do not survive to adulthood.

Ccny is also reported to be necessary for A-P patterning and organizer-specific gene expression in *Xenopus* embryos. This involves activating Cdk14 to phosphorylate Ser1490 of the PPPSP motif in the Lrp6 co-receptor of Wnt/β-catenin signaling[24,35,59]. We also constructed *ccny* and *ccnyl1* mutants. While maternal-zygotic *ccnyl1* (MZ*ccnyl1*) embryos develop normally, MZ*ccnyl1;Zccny* embryos die around 7 dpf, exhibiting a flat swim bladder and extensive apoptosis in the liver or foregut region. Therefore, the role of maternally expressed Cdk17 and Ccny remains

unclear. For the rest of the CMGC family, membrane-associated GSK3β is known to phosphorylate the PPPSP motif in Lrp6 coreceptor for Wnt/β-catenin signal activation[18]. In contrast, cytosolic GSK3β promotes the degradation of β-catenin and signal turn-off. The in vivo function of GSK3β in Hwa phosphorylation and signal activation is challenging to evaluate directly. For these reasons, we used Cdk16[DN] to compete with responsible kinases and disrupt the phosphorylation at Ser168 of Hwa. This disruption led to ventralized phenotypes (Fig. 6, Supplementary Figs. 9 and 10), confirming the importance of Cdk16, Cdk2, and GSK3β both in HEK293T cells and zebrafish embryos. However, we cannot rule out contributions from other kinases. It is reasonable that Hwa acts as a naturally occurring constitutively active receptor, like Lrp6ΔN[20]. Multiple kinases are leveraged in ensuring the axis induction process. Looking forward, the phosphorylation switch mechanism of the Hwa protein may be a general phenomenon. This opens new avenues for research on this master protein in axis induction, as well as the interplay between cell fate determination, other kinases, and signaling pathways in developmental redundancy.

## Methods

### Zebrafish and embryos
Embryos were raised at 28.5 °C and staged as previously described[60]. Fish maintenance and breeding followed the institutional animal care and use committee (IACUC) protocol, with approval by the Animal Care and Use Committee of West China Hospital, Sichuan University (NO.20220422003). To genotype *hwa*[tsu01sm] mutants with PCR, the forward primer for the mutant allele (5′-CGTGCAATCGAGCGAACTTT-3′) and wild-type allele (5′-TAGCCAACACAAGTCCTCAT-3′) were used in combination with a common reverse primer (5′-CCAGCTGCGA-CATTTCATCACAA-3′), which produced a 479 or 382 bp fragment, respectively. To construct the S168A mutation line, gRNA targeting Ser168 (100 pg), Cas9 mRNA (400 pg) and a 48 nt single-strand donor DNA oligo (10 pg) were coinjected at the 1-cell stage. Precisely edited F0 embryos were characterized by Sanger sequencing with forward 5′-ATGTTTCGGTTTCGGAGCCA-3′ and reverse 5′-AATGACATATTAGG ACCCTACCCC-3′ primers. Capped wild-type and mutant *hwa* mRNAs were synthesized with mMESSAGE mMACHINE™ SP6 (AM1340, Thermo Fisher), purified with the RNeasy Mini kit (74104, Qiagen) according to the manufacturers' instructions, and injected into the yolk at the 1-cell stage. For inhibitor treatment, zebrafish embryos were cultured with Holtfreter's solution with AZD5438 (2 μM, 10 μM) from 2 hpf to 4 hpf.

### Whole-mount in situ hybridization (WISH) and immunofluorescence (IF)
WISH was performed following the general protocol for zebrafish embryos. The linearized plasmids or PCR-amplified DNA fragments were used as templates for in vitro synthesis of digoxigenin-UTP-labeled antisense RNA probes. After WISH, embryos were fixed with 4% PFA and immersed in glycerol for photographing under a Nikon stereomicroscope (SMZ18).

### Plasmid constructs
HA and Flag-tagged wild-type Hwa expression plasmids, HA-tagged TNKS1, HA-tagged Axin1 (mouse), Flag-tagged GSK3β (human), pRL-CMV and SuperTop Flash reporter plasmids were previously reported[15]. Zebrafish *cdk2/14/16* and *ccny/ccnyl1* were cloned and constructed with a Myc or HA tag to the N-terminal or C-terminal of the coding sequence, respectively. Deletion or point mutations were introduced into the expression plasmids via a PCR-based point mutation strategy. HA-tagged Hwa (amphioxus, sea squirt and frog) expression plasmids were gifts from Xuechen Zhu, Peking University.

### Antibodies and reagents
The following antibodies and reagents were used: anti-Flag (F1804, Sigma), anti-Myc (AE010, ABclonal; sc-40, Santa Cruz), anti-HA antibody (sc-7392, Santa Cruz; AE008, ABclonal), anti-β-catenin (8480S, CST), anti-α-tubulin (T6199, Sigma; AC008, ABclonal), anti-β-actin (AC026, ABclonal), anti-His (AE003, ABclonal), anti-GST (AF5063B, Beyotime Biotechnology), Cycloheximide (HY-12320, MCE), AZD5438 (S2621, Selleck), AT7519 (S1524, Selleck), kinase buffer (#9802, CST), ATP (P0756S, NEB), protein A/G agarose (P2055, Beyotime Biotechnology), *E.coli* derived His-CDK2 (human) protein (230-00574-100, Ray Biotech), recombinant human His-GSK3β protein (HY-P74114, MCE) and human PCTAIRE1/CDK16 with CCNY protein (ab177586, Abcam) derived from Sf9 cells. GST and GST-Hwa (ΔN46) protein were produced from *E. coli* as previously described[15]. Anti-pHwa(Ser168) antiserum/antibody was produced by Shanghai Genomics in rabbits with synthetic phosphorylated peptides (VNTVPPN(p) SPVLR) and purified by HUABIO, Hangzhou. To specifically detect phosphorylation at Ser168 of the Hwa protein, the competitive non-phopeptide (VNTVPPNSPVLRV or VNTVPPNSPVLR) (Sangon Biotech) was added to the antibody solution at a final concentration of 10 μg/ml.

### Cell culture, immunoblotting and coimmunoprecipitation
HEK293T cells were cultured in high-glucose DMEM supplemented with 10% FBS and penicillin/streptomycin in a 37 °C humidified incubator with 5% CO2[61]. Plasmids were transfected with VigoFect (T001, Vigorous) or Lipo8000™ (C0533, Beyotime) according to the manufacturers' instructions. For the SuperTop Flash reporter assay, the pRL-CMV plasmid was cotransfected as the internal control. Treatment with the CDK inhibitors AZD5438 and AT7519 was performed as follows: transfected cells were cultured in prewarmed DMEM containing the desired dose of inhibitors from 24 hour posttransfection and harvested at 30 hour posttransfection. Cells were lysed with TNE buffer (10 mM Tri-HCl, pH 7.4, 150 mM NaCl, 5 mM EDTA, 1% Triton X-100) supplemented with protease and phosphatase inhibitors (B14001 & B15001, Bimake). To investigate the activation of β-catenin signaling, cytosolic β-catenin protein was extracted using a digitonin buffer (0.15 mg/ml digitonin in PBS, pH 7.2), with α-tubulin or β-actin serving as loading controls. Immunoblotting and coimmunoprecipitation were performed. Band intensity was assessed using ImageJ software.

### Reverse transcription and quantitative PCR (RT-qPCR)
WT and treated zebrafish embryos were harvested at desired stages. Total RNA was extracted with the RNeasy® Mini Kit (74104, Qiagen), and then NovoScript® 1st Strand cDNA Synthesis SuperMix (E047-01B, Novoprotein) was used to obtain cDNA. RT-qPCR was performed under standard condition with following primers: *hwa*, 5′-GCAT-CATCCCACAGGAGAAC-3′ and 5′-GTGACGTAACTTGGGTCGTA-3′; *boz*, 5′-CTTATGCCGTAGCCGGTTGT-3′ and 5′-GTTTGTCAGCG-CAGGTTGTC-3′; *chd*, 5′-TTATCCGGTTGCTCCTTCGG-3′ and 5′-GACCTCCTTCCTCCCAGAGT-3′; *vox*, 5′-CAGCTCAGGTTACGCCAAGA-3′ and 5′- TTTGTCGATCTGTTCCGGGG-3′; *gsc*, 5′-GAGACGA-CACCGAACCATTT-3′ and 5′-CCTCTGACGACGACCTTTTC-3′; *bmp2b*, 5′-CTTAGGAGACGACGGGAACG-3′ and 5′-CGGTCGATCTCGGGAAT-GAG-3′; *vent*, 5′-TTCAGAACCGGCGGATGAAG-3′ and 5′-GTAGTACCC-CACGCTTTGGT-3′; *eif4g2a*, 5′-GAGATGTATGCCACTGATGAT-3′ and 5′-GCGCAGTAACATTCCTTTAG-3′.

### Phosphorylation Validation by Mass Spectrometry (MS)
To validate phosphorylation of Hwa protein at Ser168, synthetic non-phosphorylated (VNTVPPNSPVLR) and phosphorylated peptides (VNTVPPN(p)SPVLR) were used as standards. Hwa protein was expressed in zebrafish embryos (with *hwa*-HA/*hwa*-Flag mRNA injected at the 1-cell stage in zebrafish embryos and collected at 4 hpf) and HEK293T cells (with pCS2-Hwa-HA or pCS2-Hwa-Flag plasmid transfection). Immunoprecipitated Hwa protein was then subjected to LC-MS/MS, with each condition in biological duplicate. Mass

spectrometry was performed at Tsinghua University Protein Technology Center. Protein bands were digested in-gel for MS analysis. Proteins were reduced with 25 mM dithiothreitol (DTT), alkylated with 55 mM iodoacetamide, and digested overnight at 37 °C with trypsin in 50 mM ammonium bicarbonate. Peptides were extracted twice with 1% trifluoroacetic acid in 50% acetonitrile for 30 min each, then concentrated using SpeedVac. LC-MS/MS analysis was performed on a Thermo-Dionex Ultimate 3000 HPLC system with a 60-min gradient elution at 0.300 µL/min, coupled to a Thermo Orbitrap Fusion MS. Peptide separation was achieved using a custom-packed C18 resin (300 A, 5 µm; Varian, Lexington, MA) fused silica capillary column (75 µm ID, 150 mm length; Upchurch, Oak Harbor, WA). Mobile phases were 0.1% formic acid (A) and 100% acetonitrile with 0.1% formic acid (B). The Orbitrap Fusion operated in data-dependent acquisition mode controlled by Xcalibur 3.0. Full-scan MS spectra (350-1550 m/z, 120,000 resolution) were followed by 3-s data-dependent MS/MS with 30% HCD collision energy. MS/MS spectra were analyzed using Proteome Discoverer (v1.4), with phosphorylation site localization by phosphoRS 3.1, and all MS/MS spectra of phosphorylated peptides were manually verified.

### In vitro phosphorylation and GST pull-down assay

To check whether the Ser168 phosphorylation of Hwa by kinase proteins is direct or not, purified GST-Hwa(ΔN46), recombinant His-tagged human GSK3β protein and PCTAIRE1/CDK16 with CCNY protein from Sf9 cells, and His-tagged human CDK2 protein from *E.coli* was used for in vitro analysis. For the in vitro phosphorylation assay, about 1 µg GST-Hwa(ΔN46) and His-GSK3β, CDK16-CCNY or His-CDK2 proteins were added to 30 µl kinase reaction buffer, with or without ATP (200 µM), and incubated at 37 °C for 2 h. Then, 6 µl SDS-PAGE sample buffer (5×) was added to the mixture and heated at 95 °C for 5 min before western blot analysis. For GST pull-down, 1 µg His-CDK2 protein and GST or GST-Hwa(ΔN46) protein were incubated in 500 µl TNE buffer in a 1.5 ml EP tube, with 0.5 µg anti-GST antibody each tube for 1 h. Then protein A/G agarose beads were added and further incubated for 2 h to complete the pull-down assay. Immunoblotting and Coomassie brilliant blue staining were used to quantify the phosphorylated Hwa (pHwa) and input proteins.

### Statistics and Reproducibility

All immunoblotting and RT-qPCR experiments were repeated at least three times (N ≥ 3) with the individual data points shown in the bar graphs plotted by GraphPad Prism 8. Data for statistical analysis are presented as the mean ± S.D. The significance of differences between treatments were analyzed using a two-tailed unpaired *t*-test without special mentions (Figs. 2c, d, g, 4g–j, n–p, 5b, d, g, i, 6c, 7d and Supplementary Fig. 1b, 1d, 2b, 2d, 2f, 4c, d, 5b, 5d, 5f, 7b, 7d, 7f, 7i, 11c, 11f). For microinjection and treatment of embryos, results from repeated experiments (N ≥ 2) were summed to show the total number of each treatment (n). The effects of different treatments were compared using Fisher's exact test in GraphPad Prism (all phenotypes were divided into two groups: Unchanged [Class I] and Changed [Class II-V or Class II–IV]) (Figs. 1c, f, 2a, 5e, f, 5h, j, 6e–g, and Supplementary Figs. 9a, b, 10a). Significance levels are indicated by ns ≥ 0.05, *p < 0.05, **p < 0.01, ***p < 0.001, and ****p < 0.0001, with the p-value shown in the individual figures. The sample size (n) and number of replications (N) of the experiment are described in the legend of each figure.

### Reporting summary

Further information on research design is available in the Nature Portfolio Reporting Summary linked to this article.

## Data availability

All data are included in this manuscript. Source data are provided with this paper.

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

## Acknowledgements

We would like to thank Dr. Ran Lu, Yanqiu Gao and other members of the Laboratory of Pediatric Surgery, West China Hospital, Sichuan University for their generous help in methodology and reagents. We also would like to extend our sincere thanks to Dr. Weimin Shen, Dr. Lin Zhang, Dr. Tong Lyu, Xin Liu, and Weiying Zhang for their continued support in materials and experiments. We are also grateful for the LC-MS/MS technical support from Dr. Haiteng Deng and Meng Han, Xianbing Meng in the Center of Biomedical Analysis, Tsinghua University. We also appreciate Professor Lin Li (Shanghai Institutes for Biological Sciences, CAS), Yu Rao (Tsinghua University), Xuechen Zhu (Peking University), Qiang Wang (South China University of Technology), Chengtian Zhao (Ocean University of China), Qinghua Tao (Tsinghua University), Kui Wang (Sichuan University) and Dong Deng (Sichuan University) for their kind help with plasmids, reagents and suggestions. We would like to offer our special thanks to Dr. Changxin Ma and Dr. Lei Gao for their unyielding support and extensive discussions. This research is supported by the National Natural Science Foundation of China (32170813, 31871449 and 32470853 to J.C.), the National Key R&D Program of China (2022YFC2703700, 2022YFC2703704 to J.C.), the Science and Technology Department of Sichuan (2024NSFSC0651 to J.C.) and the 1·3·5 project for disciplines of excellence–Clinical Research Fund, West China Hospital, Sichuan University (2024HXFH035 to J.C. & ZYGD23026 to X.M.).

## Author contributions

Yao. L. and Y.Y. contributed to the biochemical/zebrafish experiments, data analysis and revision of the manuscript; J.S. and Z.W. carried out microinjection and other experiments in zebrafish embryos; Q.Z., H.Z., M.L., Yaohui. L., Y.W., W.C. helped to carry out experiments; B.G., S.Q., X.M., A.M., and B.X. contributed to the experimental design and extensive discussions; J.C. conceived the project, identified the phosphorylation site, analyzed the data and wrote the manuscript.

## Competing interests

The authors declare no competing interests.
