## [Peer Review file · Nature Communications]

A Huluwa phosphorylation switch regulates embryonic axis induction

Corresponding Author: Professor Jing Chen

Version 0:

Reviewer comments:

Reviewer #1

(Remarks to the Author)

The authors previously reported that Hwa acts as a dorsal determinant by activating β -catenin signaling in zebrafish and *Xenopus*. In this study, they found that Ser168 in the PPNSP motif is required for the Hwa's axis-inducing activity in zebrafish embryos. By using biochemical studies in human culture cells, the Ser168 site was phosphorylated by several kinases, including Cdk16, Cdk2, and GSK3 β . Overexpression of dominant-negative Cdk16 reduced the Ser168 phosphorylation in human cell culture and affect axis formation in zebrafish embryos. They also generated zebrafish mutants in which Ser168 was substituted to Ala using genome-editing and found that this mutation is loss-of-function. Based on these findings, the authors concluded that Ser168 acts as a phosphorylation switch in Hwa/ β -catenin signaling for embryonic axis induction.

The authors' hypothesis is potentially interesting. However, the authors examined the biochemical mechanisms of Hwa regulation in mammalian cells but not in zebrafish, although Hwa gene exists in zebrafish genome but not in mammal. Connection between zebrafish data and mammalian cell data is insufficient. I think that there are several points which need to be addressed.

- 1) The authors mainly examined the phosphorylation and biochemical activity of Hwa in HEK293 cells (human cells). But Hwa is not conserved in human. Therefore, the authors should examine this in animals which express endogenous Hwa (e.g. zebrafish and frog). It is not difficult to perform biochemical assays in *Xenopus* embryos. It is also easy to detect Ser168 phosphorylation in zebrafish (or *Xenopus*) using pHwa antibody. The authors can confirm that overexpression and inhibition of Cdk2/16 and GSK3 increase and decrease pHwa levels, respectively, in zebrafish embryos. This experiment will strengthen the authors' hypothesis.
- 2) In Cdk2 experiments, the author did not add cyclins. Why? Related to this, the effect of Cdk2 on pHwa was not significant (Fig S6D, 5A, 5F). AZD5438 also inhibited in the absence of exogenous Cdk2 (Fig S6E). These data are not convicting.
- 3) In several western blotting data, the authors showed the band intensity. But, in such quantification, repetition is required. At least, triplicate assay is essential.
- 4) In Fig 6F experiment, some controls are required. I understand that pHwa antibody injection affect axis formation, but it might be due to the effects of injection itself, the antibody non-specific effects, or the Hwa phosphorylation-independent effects. In addition, the authors should confirm that pHwa antibody injection reduce the endogenous pHwa levels.
- 5) In Cdk-DN overexpression experiments (Fig 6, S9, S10), the authors should confirm that Cdk-DN does not affect axis formation in hwa mutant (in the absence of Hwa).
- 6) A mutant line with a serine-to-alanine (S168A) substitution is an important tool for demonstrating the authors' hypothesis. But it is difficult to understand why the authors used tsu01sm allele and why they did not use S168A homo mutants. It seems to be easier to compare the levels of hwa mRNA in wild-type embryos with those in S168A homo-mutant embryos.
- 7) Mutation introduction sometimes affect the stability of proteins. Therefore, comparison of protein levels between wild-type and mutant is more important than comparison of transcript levels. I strongly recommend the authors to confirm that S168A mutation does not affect the Hwa protein levels in "zebrafish". This confirmation is required for concluding that S168A mutation is a loss-of-function allele of Hwa in zebrafish.
- 8) Almost all graph data lacks statistical analyses. Statistics are essential.
- 9) Overall, this study lacks confirmation about kinase-mediated Hwa phosphorylation and regulation in "zebrafish". These data are provided using human cell studies although human does not have Hwa gene. Therefore, the authors data does not support the authors conclusion "A phosphorylation switch of Huluwa receptor for embryonic axis induction".

Minor points

- 1) The authors stated they performed a systematic screen in Abstract and introduction. But their experiment is “candidate approach”, but not systematic screen. The authors should rephrase this misleading words.
- 2) Hwa(S168A) is a phosphor-deficient mutant. Therefore, this mutant should be used in Fig 2B, in which only the phospho-mimicking mutants were used.
- 3) It is difficult to understand Fig. S2A. Both pho-peptide and non-pho-peptide seems to block the phosphor-state of Hwa. What does this mean?
- 4) The bottom panel in Fig S2B is strange. In regular case, phosphorylated proteins migrate slower than non-phosphor proteins in SDS-PAGE gel. As expected, in middle panel, PPase-treated Hwa protein migrated faster. However, in bottom panel, PPase-treated Hwa protein migrated slower. The authors should explain this discrepancy.
- 5) Fig S2C data indicates that anti-pHwa recognizes not only phosphor-Hwa but also non-phospho-Hwa proteins. The authors should perform both the affinity purification with phosphor-peptides and the absorption with non-phosphor-peptides before using this antibody. Related to this, the authors’ statement “Additionally, the pHwa antibody identified a specific band in Hwa(WT) and Hwa(S168E) samples but not in Hwa(Δ PPNSP) and Hwa(S168A) mutants (Fig. S2C)” in page 9 is overstatement.
- 6) The authors should explain why they focused firstly on Cdk14 and 16 among Cdks (page 10).
- 7) In Fig S4, to conclude that Cdk16, but not Cdk14, elevates pHwa, the positive controls are required. The phosphorylation of some known Cdk14 substrates by Cdk14 overexpression should be shown as a positive control.
- 8) In some Figures, the kinase overexpression induced the migration-shift of Hwa proteins on SDS-PAGE. But, in other Figures, this did not induce the migration-shift. What causes this difference?
- 9) The authors stated “Consistently, zebrafish Ccny protein recruited Cdk16 to the plasma membrane in both zebrafish embryos and HEK293T cells (Fig. S5)” in page 10. Why consistently? I feel the description is insufficient.
- 10) There are typos. For example, pWha in Fig S6a may be pHwa.
- 11) S6A in line 200 in page 10 may be S6B. Confirm the numbering of Figures again.
- 12) In page 11, the authors should explain why they used membrane-associated GSK3b construct. This construct suddenly appeared.
- 13) In Fig. 5D, why did the authors inject CDK16/CCNY proteins without hwa? I feel that they should show the data of injection of CDK16/CCNY proteins with hwa, too.
- 14) Although the authors stated “Ser168 acts as a phosphorylation switch in Hwa/ β -catenin signaling for embryonic axis induction, involving multiple kinases to ensure developmental robustness”, the words “involving multiple kinases to ensure developmental robustness” is extraordinary. The authors did not show the multiple kinases contribute the phosphorylation of endogenous Hwa and did not provide the evidence demonstrating that they ensure developmental robustness. Therefore, these words should be eliminated.

Reviewer #2

(Remarks to the Author)

I am very enthusiastic about this paper on the molecular mechanism of action of the master regulator of axial induction Huluwa. It is truly remarkable in zebrafish 100% of ventralized embryos can be obtained in such large numbers. The authors provide very convincing evidence that Ser168 is essential for function. The figures are beautiful. The mechanistic evidence is profuse, and no additional experiments are recommended, in fact some material could be removed.

Main Comments:

- 1 . I do not think it is appropriate to call Hwa a receptor, and doing this will confuse the field. Hwa is a transmembrane protein yet its extracellular domain is tiny and can be exchanged for other peptides without affecting function. We normally think of receptors as transducers of extracellular signals. I recommend omitting “receptor” from the title, line 144, and elsewhere. The use of “phosphorylation switch” is appropriate and implies that the activity of Hwa is a regulated. I realize the authors think that the situation is analogous to that of LRP6 Δ tailN, which is constitutively active but still regulated intracellularly by phosphorylation, and this is a very nice hypothesis.
- 2 . Supplementary Figure S8 using the CDK inhibitor AZD5438 is quite spectacular and a novel method of inducing dorsalization in embryos. It will be lost in the supplement and in fact goes against the main theme of the paper. I suggest removing this figure and publishing it separately as it would merit a short paper discussing AZD5438 late effects on Wnt or BMP signaling. Here it would be enough to start by saying they made a CDK16 dominant-negative.
- 3 . This work reminds us how accurate the landmark work by Davidson et al. (ref 24) was when they showed the important role of Cyclin Y, which is a membrane-bound activator of Cdks. Perhaps the role of enhanced Wnt signaling in G2/M phase could be highlighted in the discussion.

Minor Comments:

- 4 . The mass spectrometry data is difficult to follow for the uninitiated like myself, if possible help us with arrows pointing to the relevant phosphorylated peptides.
- 5 . Membrane-bound Cyclin Y should be mentioned in the Abstract as it is crucial.
- 6 . lines 134-135, please describe the assay used for Axin degradation. Was cycloheximide used?
- 7 . line 185, “beposphorylated” needs a space.
- 8 . line 189, the sentence “This enhancement was...” should be placed in line 187.
- 9 . line 193, provide full name for GST.
- 10 . line 198, inhibited instead of vanish; cite Figure 4C write after “failed to phosphorylate Hwa”.
- 11 . lines 204-209, could be explain more clearly.
- 12 . lines 289-290, a period is missing.

- 13 . line 438, gifts from Anming Meng is an important author.
- 14 . Figure 2B explain that your using short and long exposures.
- 15 . Figure 4A reads pWha instead of pHwa.
- 16 . Figure 6D, it would be better to separate to right panel (statistics) from the in-situ images.
- 17 . Figure 7G, Hwa genotypes too small to read.

Reviewer #3

(Remarks to the Author)

This manuscript provides compelling results showing the importance of phosphorylation of Serine 168 in the Huluwa (Hwa) protein, a new component of B-Catenin signaling that the authors previously showed acts as a dorsal determinant in axis formation in zebrafish and *Xenopus*. The authors new results reveal in loss and gain of function studies in zebrafish, as well as biochemical studies in HEK293 cells that Cdk16, Cdk2, and GSK3B can phosphorylate Hwa S168 and that this phosphorylation is critical for its dorsal axis induction function. They further show that Ccn1 potentiates Cdk16 activation, likely through its known role in phosphorylating and activating Cdk16. Interestingly, the Wnt receptor Lrp6 contains the same PPPSP motif that is phosphorylated by GSK3B. Lastly, they make a knock-in allele mutating S168 to alanine in the zebrafish, demonstrating that it is essential to axis formation, behaving the same as a null allele. This is exciting new work deciphering the regulatory mechanism of Hwa function, a novel protein key to dorsal axis formation. Prior to publication, several points below should be addressed.

1. In vitro transcribed mRNA can have variable capping efficiencies, which can lead to variable activity from one batch of mRNA to the next, making it difficult to determine activity levels based on mRNA amount injected. For Hwa mutated mRNAs in Fig 1 that are deemed to have reduced activity, quantitative western blots should be performed to ensure the same amount of protein is made. Also some mutant Hwa forms could generate unstable proteins, rather than affecting activity perse.
2. It seems that the Non-phosphorylated peptide also competes for the Hwa phospho-antibody in Fig S2A. Quantitation would clarify this.
3. In S2B, what does weak and strong refer to, and the numbers above the strong blot? Assuming the numbers above the blot are quantitation (are they normalized to anti-Flag?), the phosphatase inhibitor seems modestly effective. To show significance in the difference, additional biological replicates and statistics would strengthen the point.
4. Why can CDK16 + Ccn1 rescue Hwa mutants to V1 in Fig 5D? This is surprising.
5. The increase in rescue by injecting Hwa + GSKB or hwa and cdk2 is mild. Is the difference to Hwa injection alone in Fig 5E,F statistically significant? The modest rescue may reflect the presence of the endogenous activating kinase's in the embryo. An increase in chordin expression is clear though in Fig 5G.
6. Why does the ccn1 + 100 pg cdk2-DN attenuate axis formation equally or better to 200 pg cdk2-DN? Please clarify the logic behind adding the ccn1 here?
7. Excellent n values are provided for most experiments, however, N values should also be included for the number of biological replicates. At least 2 biological replicates should be performed, including for the western blots, and the number of such replicates should be included.
8. A typo in Fig 4A, pWha.
9. What stage are the embryos in Fig S5?

Version 1:

Reviewer comments:

Reviewer #1

(Remarks to the Author)

The authors have responded appropriately to most of the comments.

Reviewer #2

(Remarks to the Author)

This paper should be accepted

Reviewer #3

(Remarks to the Author)

In this revision the authors have greatly strengthened the conclusions of their manuscript, by providing additional experiments, controls, quantitation, and clarifications. The work using in vivo studies in the zebrafish and in vitro work in cell culture reveals an important new regulatory mechanism of Huluwa, a novel protein key to dorsal axis formation in the B-Catenin pathway, phosphorylation of Ser168 of Huluwa by Cdk16, which in turn is activated by Ccn1.

A point-by-point Response to Reviewers

"A phosphorylation switch of Hwuwa receptor for embryonic axis induction" by Yao Li et al.

We sincerely appreciate the time taken by the reviewers in assessing our manuscript and for providing valuable and constructive comments. We have followed their suggestions and have addressed all the concerns with additional experiments. We believe that the revisions have improved the manuscript and strengthened the conclusion. In response to the reviewers' comments, we conducted *in vivo* experiments in embryos to confirm the phosphorylation of Hwa at Ser168 in zebrafish, and also repeated the microinjection experiments and immunoblotting for statistical analysis, performed *in vitro* phosphorylation experiments with purified Hwa and CDK2/CDK16/GSK3 β proteins, and corrected inaccurate statements and typos. We present these new data in the figures and supplementary figures in a significantly revised manuscript. Changes in the manuscript are highlighted in red. We are providing a point-by-point response as follows:

Reviewer #1 (Remarks to the Author):

The authors previously reported that Hwa acts as a dorsal determinant by activating β -catenin signaling in zebrafish and *Xenopus*. In this study, they found that Ser168 in the PPNSP motif is required for the Hwa's axis-inducing activity in zebrafish embryos. By using biochemical studies in human culture cells, the Ser168 site was phosphorylated by several kinases, including Cdk16, Cdk2, and GSK3 β . Overexpression of dominant-negative Cdk16 reduced the Ser168 phosphorylation in human cell culture and affect axis formation in zebrafish embryos. They also generated zebrafish mutants in which Ser168 was substituted to Ala using genome-editing and found that this mutation is loss-of-function. Based on these findings, the authors concluded that Ser168 acts as a phosphorylation switch in Hwa/ β -catenin signaling for embryonic axis induction.

The authors' hypothesis is potentially interesting. However, the authors examined the biochemical mechanisms of Hwa regulation in mammalian cells but not in zebrafish, although Hwa gene exists in zebrafish genome but not in mammal. Connection between zebrafish data and mammalian cell data is insufficient. I think that there are several points which need to be addressed.

Response: Thank you very much; we sincerely appreciate both your positive assessment of our work and your valuable suggestions.

1) The authors mainly examined the phosphorylation and biochemical activity of Hwa in HEK293 cells (human cells). But Hwa is not conserved in human. Therefore, the authors should examine this in animals which express endogenous Hwa (e.g. zebrafish and frog). It is not difficult to perform biochemical assays in *Xenopus* embryos. It is also easy to detect Ser168 phosphorylation in zebrafish (or *Xenopus*) using pHwa antibody. The authors can confirm that overexpression and inhibition of Cdk2/16 and GSK3 increase and decrease pHwa levels, respectively, in zebrafish embryos. This experiment will strengthen the authors' hypothesis.

Response: Thank you for your constructive suggestions. We have followed your advice and conducted *in vivo* experiments in zebrafish embryos to verify the phosphorylation mechanism found in mammalian cells. Although it is not difficult to perform biochemical assays in frog embryos, a similar investigation of Hwa in zebrafish poses several challenges.

Firstly, in the frog, the yolk is present in granules/particles, most of which do not dissolve in extraction buffer with NaCl concentration below 150 mM, with little effect on protein extraction and immunoblotting for early-stage embryos. However, the yolk in early-stage zebrafish embryos is readily soluble and is in a large amount, which has dramatic effects on the performance of SDS-PAGE and introduces non-specific signals in immunoblotting. To overcome this problem, we performed a deyolking step to minimize the interference of non-specific signals caused by yolk proteins, but this, in turn, may lead to the loss of target proteins.

Secondly, endogenous Hwa levels are very low in zebrafish, and cannot even be detected by mass spectrometry in early-stage embryos (**Table S1. Nuclei and whole-cell proteome, related to Figure 1** from Weimin Shen et al., *Cell*, 2022, 185[26]:4954-

4970.e20). It was also not possible to detect endogenous Hwa/pHwa in embryos with immunoblotting, despite the good specificity of the pHwa antibody for overexpressed Hwa as indicated in **Supplementary Fig. 1**. Therefore, we conducted overexpression of Hwa in embryos to allow detection of Ser168 phosphorylation in zebrafish. Consistent with the results from mammalian cells, reaction with the pHwa resulted in reliable, although not strong, signals in embryos expressing wild-type Hwa protein, but not Hwa(S168A) (**Fig. 5a-b**).

Fig. 5a-d. Ser168 of Hwa is phosphorylated in zebrafish (a) Immunoblotting of pHwa in zebrafish embryos injected with 200 pg WT or S168A mutant of *hwa*-HA mRNA at the 1-cell stage and harvested at 4 hpf. (b) Quantifications of relative pHwa levels in embryos treated as in (a), N=6. N, number of biological replicates; Significant difference is indicated by **** $p < 0.0001$, with the p-value illustrated.

Your advice on the manipulation of kinases to confirm their contribution to Ser168 phosphorylation in zebrafish is valuable and constructive, and we have followed your advice and performed additional kinase manipulation. As endogenous Gsk3 α/β and dozens of Cdks are highly expressed in early-stage embryos, it is very challenging to detect increases in pHwa when Cdk2/Cdk16/ GSK3 β are overexpressed. After arduous efforts with much trial and error, we finally obtained reliable and reproducible results. The immunoblotting results indicated that both *cdk2+ccna2* and *cdk16+ccnyl1* co-expression reduced the total Hwa protein level (**Response Fig. 1a**), indicating that kinase overexpression reduced Hwa's stability. This made it more difficult to monitor the pHwa level. As the Ccnyl1-HA and Hwa-HA bands were almost overlapping, we chose to use *cdk2+ccna2* for pHwa quantification. The *cdk2+ccna2* combination decreased the level of total Hwa, while significantly increasing the relative level of pHwa in zebrafish (**Response Fig. 1b-c**); this is consistent with the results seen in HEK293T cells (**Response Fig. 2**).

Response Fig. 1 Co-expression of Cdk2/Ccna2 increased the relative level of pHwa in zebrafish embryos. (a) Immunoblotting of embryos injected with 200 pg *hwa*-HA mRNA combined with different kinase-cyclin combinations. (b) Immunoblotting of pHwa in embryos injected with 200 pg *hwa*-HA mRNA combined with *Flag-gfp* (200 pg) or *Myc-cdk2/ccna2*-HA (100 pg + 100 pg) mRNAs. (c) Quantifications of relative pHwa levels in embryos treated as in (b), N=4. Total Hwa proteins were used as reference for pHwa quantification. N, number of biological replicates; Significant difference is indicated by ** $p < 0.01$, with the p-value illustrated.

Response Fig. 2 Co-expression of Cdk2/Ccna2 increased pHwa levels in HEK293T cells. (a) Immunoblotting of pHwa from HEK293T cells transfected with Hwa-Flag with different Myc-Cdk2, alone or combined with Ccna2-HA. (b) Quantifications of relative pHwa levels in HEK293T cells treated as in (a), N=4. Total Hwa proteins were used as loading controls in immunoblotting. N, number of biological replicates; Significant differences are indicated by ns ≥ 0.05 , * $p < 0.05$, and ** $p < 0.01$, with individual p-values shown.

Moreover, we attempted to reduce the activities of endogenous kinases using a commercial inhibitor, AZD5438, which is a well-studied pan CDK and GSK3 inhibitor (Gongyu Shi et al., AZD5438 a GSK-3a/b and CDK inhibitor is antiapoptotic modulates mitochondrial activity and protects human neurons from mitochondrial toxins. *Sci Rep.* 2023, 23;13[1]:8334). Consistent with the results from human cells, inhibition of CDKs and GSK3 β with AZD5438 dramatically decreased the pHwa level in zebrafish (**Fig. 5c-d**), confirming the contribution of Cdk2/16 and GSK3 β to the phosphorylation of Hwa at Ser168 in zebrafish embryos.

Fig. 5c-d. Ser168 of Hwa is phosphorylated in zebrafish (c) Immunoblotting of pHwa in zebrafish injected with 200 pg *hwa*-HA mRNA at the 1-cell stage, followed by treatment with AZD5438 from 2-4 hpf. (d) Quantifications of relative pHwa levels in embryos treated as in (c), N=4. N, number of biological replicates; Significant differences between treatments are indicated by ns \geq 0.05, *p < 0.05, **p < 0.01, ***p < 0.001, and ****p < 0.0001, with individual p-values illustrated.

Besides, the *in vitro* biochemical experiments with purified proteins (**Fig. 4k-p**) further demonstrated that purified CDK2/16 and GSK3 β kinase can phosphorylate purified GST-Hwa protein at Ser168 directly to validate the kinase-substrate relationships.

Fig. 4k-p. Ser168 of Hwa can be phosphorylated by multiple kinases *in vitro*. (k) *In vitro* phosphorylation of purified Hwa by recombined CDK16/CCNY proteins in the absence or presence of ATP. (l) *In vitro* phosphorylation of purified Hwa by recombinant His-GSK3 β in the absence or presence of ATP. (m) *In vitro* phosphorylation of purified Hwa by recombinant His-CDK2 proteins in the absence or presence of ATP. (n-p) Quantifications of relative pHwa levels in *in vitro* phosphorylation experiments treated with different kinase proteins as in (k-m), N=3. The arrow and arrowhead indicate the Hwa and kinase proteins, respectively (k-m). Total Hwa (k-m) proteins were applied as loading controls for quantification in immunoblotting. N, number of biological replicates; Significant differences are indicated by *p < 0.05, **p < 0.01, ***p < 0.001, and ****p < 0.0001, with individual p-values shown.

2) In Cdk2 experiments, the author did not add cyclins. Why? Related to this, the effect of Cdk2 on pHwa was not significant (Fig S6D, 5A, 5F). AZD5438 also inhibited in the absence of exogenous Cdk2 (Fig S6E). These data are not convicting.

Response: Thank you for your comments. A previous study indicated that Cdk2 is auto-active even without the presence of cyclin proteins (“Consistent with a role of CDK2 in auto-activation, inhibition of CDK2 in human cells either by pharmacological inhibition of CDK2 or by the coexpression of the CDK2 inhibitors p21 or p27, inhibited CDK2 Thr-160 phosphorylation. Our results demonstrate that CDK2 is capable of autophosphorylation at Thr160.” from Tarek Abbas et al., Autocatalytic phosphorylation of CDK2 at the activating Thr160, *Cell Cycle*, 2007; 6[7]:843-52). To address your concerns, we re-performed the experiment with an expanded gradient and demonstrated that exogenous Cdk2 could indeed increase the pHwa level in a dose dependent manner (**Supplementary Fig. 7c-d**).

Supplementary Fig. 7c-d. Cdk2 phosphorylates Hwa at Ser168. (c) Immunoblotting of pHwa from HEK293T cells transfected with Hwa-Flag with different doses of Myc-Cdk2. (d) Quantifications of relative pHwa levels in HEK293T cells treated as in (c), N=3. Total Hwa proteins were used as loading controls in immunoblotting. N, number of biological replicates; Significant differences are indicated by *p < 0.05, **p < 0.01, and ***p < 0.001, with individual p-values shown.

Besides, following your suggestion, we did indeed detect a greater increase in pHwa with co-expression of Cdk2 and Cyclin A (Ccna2 in zebrafish). However, both Hwa and Cdk2 protein levels decreased in the presence of Ccna2, indicating that the phosphorylated Hwa protein is unstable and may undergo dynamic turnover. Therefore, in the main text, we have described the use of Cdk2 alone, as its effects on pHwa are reliable and significant.

Response Fig. 2 Co-expression of Cdk2/Ccna2 increased pHwa in HEK293T cells.

(a) Immunoblotting of pHwa from HEK293T cells transfected with Hwa-Flag with different Myc-Cdk2 alone or combined with Ccna2-HA. (b) Quantifications of relative pHwa levels in HEK293T cells treated as in (a), N=4. Total Hwa proteins were used as loading controls in immunoblotting. N, number of biological replicates; Significant differences are indicated by ns \geq 0.05, *p < 0.05, and **p < 0.01, with individual p-values shown.

Thank you for your suggestions, on the results from the zebrafish embryos. Previous *hwa*+*cdk2* rescue was done at a later stage (the 8-16-cell stage). To address your concern, we performed additional rescue injection at the 1-cell stage as with Cdk16/Ccnyl1, resulting in a more significant increase (Fig. 5h-j).

Fig. 5h-j. Cdk2 enhanced the axis-inducing activity of Hwa. (h) Rescue efficiency of 5 pg *hwa*-HA mRNA alone or together with *Myc*-*cdk2* injected at the 1-cell stage in *Mhwa*^{*tsu01sm/tsu01sm*} embryos, N=3. (i) Expression levels of *boz* and *chd* were quantified by RT-qPCR in embryos rescued by different mRNA combinations of *hwa*-HA and *Myc*-*cdk2* as in (h), N=3. (j) Rescue efficiency of lower dose (1.0 or 0.5 pg) of *hwa*-HA mRNA alone or together with Flag-*mGSK3β* and *Myc*-*cdk2* injected at the 1-cell stage in *Mhwa*^{*tsu01sm/tsu01sm*} embryos, N=3. Phenotypes were grouped as in (h). *ef4g2a* was the internal reference in (g, i). V, ventralized; N, normal; D, dorsalized;

V2<V1<N<D1<D2; N, number of biological replicates; n, total number of embryos in each treatment; Significant differences between treatments are indicated by ns \geq 0.05, *p < 0.05, **p < 0.01, ***p < 0.001, and ****p < 0.0001, with individual p-values shown.

In terms of “AZD5438 also inhibited in the absence of exogenous Cdk2” (update **Supplementary Fig.7e**), this was due to the presence of **endogenous** CDKs and GSK3 α/β kinases, which also contribute to pHwa.

3) In several western blotting data, the authors showed the band intensity. But, in such quantification, repetition is required. At least, triplicate assay is essential.

Response: Thank you for your constructive suggestions. We have followed your suggestions and provided both representative western blotting figures and statistical bar graphs providing illustrations of the individual data. All Immunoblotting experiments are repeated at least three times (N=3-6).

4) In Fig 6F experiment, some controls are required. I understand that pHwa antibody injection affect axis formation, but it might be due to the effects of injection itself, the antibody non-specific effects, or the Hwa phosphorylation-independent effects. In addition, the authors should confirm that pHwa antibody injection reduce the endogenous pHwa levels.

Response: Thank you for your valuable suggestions. We performed additional experiments with IgG protein as a negative control to rule out the non-specific effects of antibody injection, as updated in **Fig. 6g**.

g

Fig. 6g. Attenuating phosphorylation of Ser168 by pHwa antibody disrupts the axis-inducing activity of Hwa. (g) Phenotypes of wild-type embryos injected with pHwa antibody or IgG at different doses, N=4. N, number of biological replicates; n, total number of embryos in each treatment; Significant differences between treatments are indicated by ****p < 0.0001.

The pHwa antibody was unable to detect endogenous pHwa. Therefore, we were not able to evaluate the reduction in endogenous pHwa after antibody injection. We sincerely appreciate your understanding. On the other hand, the injected antibody may function in binding and blocking the active pSer168 site, instead of directly degrading Hwa protein. As in the TRIM-away technology, both the specific antibody and the Trim21 protein are needed to degrade and downregulate target proteins (**Ref:** Xiao Chen et al., Degradation of endogenous proteins and generation of a null-like phenotype in zebrafish using Trim-Away technology. *Genome Biol.*, 2019, 23;20[1]:19).

5) In Cdk-DN overexpression experiments (Fig 6, S9, S10), the authors should confirm that Cdk-DN does not affect axis formation in hwa mutant (in the absence of Hwa).

Response: Thank you for your suggestions. We injected Cdk2-DN and Cdk16-DN mRNA in the *hwa* mutant and observed that these two dominant negative proteins had little effect from on axis formation, with only a tiny part of the embryos showing the V1 phenotype. This further confirmed that the function of CDK-DN on axis formation is largely dependent on Hwa.

Response Fig. 2. Cdk2^{DN}/Cdk16^{DN} did not affect axis formation in hwa mutant. Phenotypes of *Mhwa* mutant embryos injected with *Myc-cdk2^{DN}* or *Myc-cdk16^{DN}*

mRNA, N=3. N, number of biological replicates; n, total number of embryos in each treatment; Significant differences between treatments was indicated by $ns \geq 0.05$.

6) A mutant line with a serine-to-alanine (S168A) substitution is an important tool for demonstrating the authors' hypothesis. But it is difficult to understand why the authors used tsu01sm allele and why they did not use S168A homo mutants. It seems to be easier to compare the levels of hwa mRNA in wild-type embryos with those in S168A homo-mutant embryos.

Response: Thank you for your helpful suggestions. Following your advice, for clarity, we have updated **Fig. 7** to use the S168A homo mutants, while using the genetic complementation data of S168A/tsu01sm for support in **Supplementary Fig.11**.

Fig. 7. The endogenous function of the S168A mutation in zebrafish embryos. (a) The CRISPR/Cas9-based editing strategy was applied to construct the S168A mutation in zebrafish embryos. The gRNA targeting site and PAM sequence are highlighted in yellow and blue, respectively; Cas9 mRNA, gRNA, and single-strand donor DNA were coinjected at the 1-cell stage. (b) Sanger sequencing of WT and *Mhwa*^{S168A/S168A} females confirmed the successful construction of the *hwa*S168A homozygous mutant line. (c) Phenotypes of *Mhwa*^{S168A/S168A} embryos at 24 hpf could be divided into two classes: both lacking an embryonic axis, and *Mhwa*^{S168A/+} embryos were all normal. (d) Marker genes were quantified by RT-qPCR at 4 hpf and 6 hpf for WT and *Mhwa*^{S168A/S168A} embryos, N=3. (e) Loss of expression of dorsal/organizer markers (*chd* and *gsc* at 6 hpf) was detected by WISH in *Mhwa*^{S168A/S168A} embryos. (f) The working model of the Hwa protein. In wild-type embryos, Hwa protein is phosphorylated at Ser168 by multiple kinases and activated, enhancing Trks1/2-mediated degradation

of the Axin protein, increasing stability of β -catenin in the cytosol. β -catenin then translocates into the nucleus and turns on downstream target genes. However, in *Mhwa*^{S168A/S168A} mutant embryos or *Mhwa*^{tsu01sm/tsu01sm} embryos supplied with Hwa(S168A) mutant protein, the point mutation switches off the activity of the Hwa protein. N, number of biological replicates; Significant differences between embryos are indicated by **p < 0.01, ***p < 0.001, and ****p < 0.0001, with individual p-values illustrated.

7) Mutation introduction sometimes affect the stability of proteins. Therefore, comparison of protein levels between wild-type and mutant is more important than comparison of transcript levels. I strongly recommend the authors to confirm that S168A mutation does not affect the Hwa protein levels in “zebrafish”. This confirmation is required for concluding that S168A mutation is a loss-of-function allele of Hwa in zebrafish.

Response: Thank you for your valuable suggestions. We followed your advice and evaluated the expression of WT and mutant Hwa in zebrafish embryos. This showed that both the wild-type and S168A mutant Hwa were expressed at similar levels after injection of the same mRNA dose (**Fig. 5a-b** shown above), indicating that the S168A mutation did not affect Hwa stability. Besides, when coinjected with Flag-GFP, both Hwa-HA(WT) and Hwa-HA (S168A) were expressed at similar levels as shown in **Supplementary Fig. 1a-b**.

Supplementary Fig. 1a-b. S168A point mutation did not affect the stability of Hwa.

(a) Immunoblotting of Hwa-HA and Flag-GFP in zebrafish embryos injected 200 pg WT or S168A mutant of *hwa*-HA mRNA with 200 pg Flag-*gfp* mRNA at the 1-cell stage and harvested at 4 hpf. (b) Quantifications of relative Hwa protein levels in embryos treated as in (a), N=4. GFP was used as references for quantification. N, number of biological replicates; Significant difference is indicated by ns \geq 0.05.

8) Almost all graph data lacks statistical analyses. Statistics are essential.

Response: Thank you for your suggestions. We followed your suggestions and have provided statistical analyses in the revised manuscript, with N (number of biological replicates), n (total number of embryos in each treatment) and the significance of differences between treatments illustrated in all figures and legends.

9) Overall, this study lacks confirmation about kinase-mediated Hwa phosphorylation and regulation in "zebrafish". These data are provided using human cell studies although human does not have Hwa gene. Therefore, the authors data does not support the authors conclusion "A phosphorylation switch of Hwa receptor for embryonic axis induction".

Response: Thank you for your valuable suggestions. We have revised the manuscript comprehensively following your suggestions. We performed numerous experiments to confirm our findings, especially experiments with *in vivo* zebrafish data to support and strengthen our conclusion that Ser168 is phosphorylated and functions as a phosphorylation switch on Hwa for axis induction in zebrafish. These new zebrafish data are presented in **Fig. 5** and **Fig. 7**.

Minor points

1) The authors stated they performed a systematic screen in Abstract and introduction. But their experiment is "candidate approach", but not systematic screen. The authors should rephrase this misleading words.

Response: Thank you for your valuable suggestions. We have rephrased the statement "systematic screen" with "candidate screening approach", as shown on Page 3, Line 31 and Page 5, Line 85 in the revised manuscript.

2) Hwa(S168A) is a phosphor-deficient mutant. Therefore, this mutant should be used in Fig 2B, in which only the phospho-mimicking mutants were used.

Response: Thank you for your valuable suggestions. We performed additional experiments with three different mutant forms, including **Hwa(S168A)**, Hwa(S168E) and Hwa(Δ PPNSP) in the revised **Fig. 2b**.

Fig. 2b-c. Mutation at Ser168 of Hwa attenuates the activation of β-catenin signaling. (b) Immunoblotting of β-catenin from the cytosol (active form) and total cell lysate (TCL) of HEK293T cells overexpressing wild-type or mutant Hwa-HA protein. PPNSP motif deletion or Ser168 mutation nearly abolished the activation of the β-catenin signal by Hwa. (c) Quantifications of relative cytosolic β-catenin levels in HEK293T cells treated as in (b), N=3. α-tubulin was used as the loading control, and relative protein levels are indicated in (c). Significant differences are indicated by **p < 0.01 and ***p < 0.001, with the p-values shown.

3) It is difficult to understand Fig. S2A. Both pho-peptide and non-pho-peptide seems to block the phosphor-state of Hwa. What does this mean?

Response: Thank you for your concern and valuable suggestions. We have followed your suggestions and performed additional experiments using purified pHwa antibody, instead of #11485 serum. We have thoroughly updated the figure with representative WB data and statistics (**Supplementary Fig. 2**) to demonstrate the specificity of the pHwa antibody. We are sorry for the confusion and appreciate your understanding.

Supplementary Fig. 2. Validation of the specificity of the pHwa antibody. (a) Immunoblotting of pHwa from samples treated with λ -PPase or λ -PPase & phosphatase inhibitor. (b) Quantifications of relative pHwa levels treated as in (a), N=4. (c) Immunoblotting of pHwa in HEK293T cells transfected with wild-type or S168A mutant of Hwa-HA. (d) Quantifications of relative pHwa levels treated as in (c), N=4. (e) Immunoblotting of pHwa in HEK293T cells transfected with wild-type or mutants of Hwa-Flag. (f) Quantifications of relative pHwa levels treated as in (e), N=4. N, number of biological replicates; Significant differences are indicated by ns ≥ 0.05 , ***p < 0.001, and ****p < 0.0001.

4) The bottom panel in Fig S2B is strange. In regular case, phosphorylated proteins migrate slower than non-phosphorylated proteins in SDS-PAGE gel. As expected, in middle panel, PPase-treated Hwa protein migrated faster. However, in bottom panel, PPase-treated Hwa protein migrated slower. The authors should explain this discrepancy.

Response: Thank you for your concern and comments. We apologize for the confusion caused by the previous presentation. We have performed additional experiments with the purified pHwa antibody. These did not show a consistent migration shift in the total Hwa protein band. Therefore, we suggest that the

description “migrated slower” did not really represent a migration shift in pHwa, but rather deformed/distorted protein bands caused by accidental errors during electrophoresis or transfer as the SDS-PAGE gel was soft and flexible. We have included the revised data in **Supplementary Fig. 2a**, as shown above.

5) Fig S2C data indicates that anti-pHwa recognizes not only phosphor-Hwa but also non-phospho-Hwa proteins. The authors should perform both the affinity purification with phosphor-peptides and the absorption with non-phosphor-peptides before using this antibody. Related to this, the authors’ statement “Additionally, the pHwa antibody identified a specific band in Hwa(WT) and Hwa(S168E) samples but not in Hwa(Δ PPNSP) and Hwa(S168A) mutants (Fig. S2C)” in page 9 is overstatement.

Response: Thank you for your concern and comments. We apologize for the confusion caused by the previous presentation. The results shown in the previous Fig. S2C were from experiments that used the #11485 serum, the specificity of which is relatively lower than that of the purified antibody. We have performed additional experiments using this purified pHwa antibody, and have updated the results in **Supplementary Fig. 2e**.

6) The authors should explain why they focused firstly on Cdk14 and 16 among Cdks (page 10).

Response: Thank you for your concern and comments. The Ser168 site is located within the intracellular PPNSP motif, which is similar to the PPPS/TP motif seen in LRP6. The first PPPSP motif (Ser1490) of Lrp6 is phosphorylated by Cdk14/Ccny in *Xenopus* (Ref24). Besides, Cdk14 and Cdk16 belong to the same subfamily (**Supplementary Fig. 8b**), and they have previously been reported to be phosphorylated and activated by membrane-targeted cyclins (Cyclin Y [Ccny]) or (Cyclin Y like 1 [Ccnyl1]). Therefore, our first focus was on Cdk14 and Cdk16 among the Cdks. We are sorry for the confusion and have provided a clearer statement in the revised manuscript on Page 10 Line 189-195.

7) In Fig S4, to conclude that Cdk16, but not Cdk14, elevates pHwa, the positive controls are required. The phosphorylation of some known Cdk14 substrates by Cdk14 overexpression should be shown as a positive control.

Response: We really appreciate your concern and suggestions, which are very helpful. A previous study suggested that the Cdk14 protein could phosphorylate Ccny (Shan Li et al., Phosphorylation of cyclin Y by CDK14 induces its ubiquitination and degradation. *FEBS Lett*, 2014, 29;588[11]:1989-96). We performed additional experiments to verify the activity of Cdk14, which phosphorylated Ccny/Ccnyl1 and generated a migration-shift band, shown by the red star in **Supplementary Fig. 4a & 4e**. However, it did not promote Hwa phosphorylation efficiently.

Supplementary Fig. 4a-d. Cdk16, but not Cdk14, elevated pHwa with Ccny/Ccnyl1.

(a) Immunoblotting of pHwa in HEK293T cells transfected with Hwa-Flag and Myc-Cdk14, in the absence or presence of Cyclin Y proteins (Ccny/Ccnyl1-HA). (b) Immunoblotting of pHwa in HEK293T cells transfected with Hwa-Flag and Myc-Cdk16, in the absence or presence of Cyclin Y (Ccny/Ccnyl1-HA). Total Hwa-Flag was applied as the reference. (c) and (d) Quantifications of relative pHwa levels in (a) and (b), respectively, N=3. (e) Immunoblotting of phosphorylated Ccny (a migration-shift band indicated by the black arrow) in HEK293T cells transfected with Ccny-HA alone or cotransfected with Myc-Cdk14. Red star indicated the migration-shift band of phosphorylated Ccny/Ccnyl1 in (a & e). Total Hwa proteins (a & b) were used as references for quantification. N, number of biological replicates; Significant differences between embryos are indicated by *p < 0.05, and **p < 0.01, with individual p-values shown.

8) In some Figures, the kinase overexpression induced the migration-shift of Hwa proteins on SDS-PAGE. But, in other Figures, this did not induce the migration-shift. What causes this difference?

Response: Thank you for your concern and comments. We apologize for the confusion in the previous presentation. We performed additional experiments and did not find a consistent migration shift of Hwa protein in different batches. Therefore, it is likely that "migration-shift" did not represent a migration shift of Hwa induced by phosphorylation, but instead indicated deformed/distorted protein bands, possibly caused by accidental errors during electrophoresis or transfer as the SDS-PAGE gel was soft and flexible. We have updated the manuscript with new representative figures.

9) The authors stated "Consistently, zebrafish Ccny protein recruited Cdk16 to the plasma membrane in both zebrafish embryos and HEK293T cells (Fig. S5)" in page 10. Why consistently? I feel the description is insufficient.

Response: Thank you for your concern and comments. We apologize for the confusion. What we wanted to express was that "**Consistent with the reported function of Ccny/Ccny1**". We have rephrased the statement on Page 11, Line 215.

10) There are typos. For example, pWha in Fig S6a may be pHwa.

Response: Thank you for your concern and suggestions. We have corrected the typos and updated the data in **Supplementary Fig 5a & 5c**.

11) S6A in line 200 in page 10 may be S6B. Confirm the numbering of Figures again.

Response: Thank you for your concern and comments. We have corrected the numbering and statement in the revised manuscript on Page 11.

12) In page 11, the authors should explain why they used membrane-associated GSK3b construct. This construct suddenly appeared.

Response: Thank you for your concern and suggestions. Cytosolic GSK3 β plays a negative role in Wnt signaling, while membrane-associated GSK3 β can activate Wnt

signaling through phosphorylation of the LRP6 co-receptor (Xin Zeng et al., A dual-kinase mechanism for Wnt co-receptor phosphorylation and activation. *Nature*, 2005, 8;438[7069]:873-7). Therefore, we used both soluble GSK3 β and membrane-associated GSK3 β to check their kinase activities in the phosphorylation of Hwa at Ser168 (**Fig. 4d, 4i, & Supplementary Fig. 7a-b**), while when checking downstream signal activation and enhancement effects on the efficiency of Hwa rescue, only membrane-associated GSK3 β was used (**Fig. 5j**). We have provided a clearer introduction and statement on this in the revised manuscript on Page 12.

13) In Fig. 5D, why did the authors inject CDK16/CCNY proteins without hwa? I feel that they should show the data of injection of CDK16/CCNY proteins with hwa, too.

Response: Thank you for your concern and suggestions. In principle, injected proteins produce more rapid and robust effects than mRNA. What we wanted to show is that even supplying CDK16/CCNY protein in *hwa* mutant embryos was not able to efficiently rescue body axis formation as further support of the mRNA-injection results. Both mRNA and protein injections demonstrated the effect of CDK16/CCNY is dependent on Hwa. Generally, protein injection is not essential for such experiments, and mRNA injection is sufficient to support the conclusion. For brevity and clarity, we have removed the protein injection data in the revised manuscript.

14) Although the authors stated "Ser168 acts as a phosphorylation switch in Hwa/ β -catenin signaling for embryonic axis induction, involving multiple kinases to ensure developmental robustness", the words "involving multiple kinases to ensure developmental robustness" is extraordinary. The authors did not show the multiple kinases contribute the phosphorylation of endogenous Hwa and did not provide the evidence demonstrating that they ensure developmental robustness. Therefore, these words should be eliminated.

Response: Thank you for your comments and suggestions. We have removed the inaccurate statements in the revised manuscript according to your advice on Page 3, Line 44.

Reviewer #2 (Remarks to the Author):

NCOMMS-24-06806: Li et al. A phosphorylation switch of Hwu receptor for embryonic axis induction

I am very enthusiastic about this paper on the molecular mechanism of action of the master regulator of axial induction Hwu. It is truly remarkable in zebrafish 100% of ventralized embryos can be obtained in such large numbers. The authors provide very convincing evidence that Ser168 is essential for function. The figures are beautiful. The mechanistic evidence is profuse, and no additional experiments are recommended, in fact some material could be removed.

Response: We sincerely appreciate your positive assessment and supportive evaluation of our work.

Main Comments:

1 . I do not think it is appropriate to call Hwu a receptor, and doing this will confuse the field. Hwu is a transmembrane protein yet its extracellular domain is tiny and can be exchanged for other peptides without affecting function. We normally think of receptors as transducers of extracellular signals. I recommend omitting "receptor" from the title, line 144, and elsewhere. The use of "phosphorylation switch" is appropriate and implies that the activity of Hwu is a regulated. I realize the authors think that the situation is analogous to that of LRP6deltaN, which is constitutively active but still regulated intracellularly by phosphorylation, and this is a very nice hypothesis.

Response: Thank you for your comments and constructive suggestions. We cannot agree with you more. We suppose that the situation with Hwu protein is analogous to that of LRP6deltaN, which is constitutively active but still regulated intracellularly by phosphorylation. However, the extracellular domain of Hwu is dispensable for its function. It thus appears that Hwu does not function as a conventional "receptor" to receive and transduce extracellular signals. Perhaps it is unconventional, but at this point, we accept your suggestions and have omitted the "receptor" description in the revised manuscript, using "protein" instead, as updated in the title and the entire manuscript.

2 . Supplementary Figure S8 using the CDK inhibitor AZD5438 is quite spectacular and a novel method of inducing dorsalization in embryos. It will be lost in the supplement

and in fact goes against the main theme of the paper. I suggest removing this figure and publishing it separately as it would merit a short paper discussing AZD5438 late effects on Wnt or BMP signaling. Here it would be enough to start by saying they made a CDK16 dominant-negative.

Response: Thank you for your valuable suggestions. We have removed this figure in the revised manuscript. As for AZD5438, it can inhibit both Cdks and GSK3 α/β , exerting multiple effects on different signaling pathways during early developmental stages. Its effect is complicated and this figure should be removed. Thank you very much.

3 . This work reminds us how accurate the landmark work by Davidson et al. (ref 24) was when they showed the important role of Cyclin Y, which is a membrane-bound activator of Cdks. Perhaps the role of enhanced Wnt signaling in G2/M phase could be highlighted in the discussion.

Response: Thank you for your valuable suggestions. We have added this part in the revised manuscript to highlight the vital role of enhanced Wnt signaling in G2/M phase during development, as shown in the discussion part, on Page 18.

Minor Comments:

4 . The mass spectrometry data is difficult to follow for the uninitiated like myself, if possible help us with arrows pointing to the relevant phosphorylated peptides.

Response: Thank you for your helpful suggestions. We have labeled the relevant fragmentation pattern with a red circle and the specific phosphorylated peptide signal with a red rectangle in Fig.3. It is obvious that the Hwa protein is expressed in both HEK293T cells (c) and zebrafish embryos (d), showing similar patterns and peaks as the synthetic phosphorylated peptide (b), but not that of the non-phosphorylated peptide (a).

Fig. 3. The phosphorylation of Hwa protein at Ser168 was validated by LC-MS/MS. The red circle indicated the specific collision-induced fragmentation to identify the phosphorylation state and the red rectangle indicated the corresponding peak in the MS profile.

5 . Membrane-bound Cyclin Y should be mentioned in the Abstract as it is crucial.

Response: Thank you for your concern and comments. We have updated and revised this statement in the Abstract.

6 . lines 134-135, please describe the assay used for Axin degradation. Was cycloheximide used?

Response: Thank you for your concern and comments. We used cycloheximide in our previous work (Ref15) , while we used gradient doses of Hwa-HA protein (WT & S168A) in the present work. We found that higher doses of wild-type Hwa reduced the level of Axin1, while the same dose of the S168A Hwa mutant did not. This demonstrated that the S168A mutation disrupted Hwa-mediated Axin1 degradation for downstream signal activation.

7 . line 185, "beposphorylated" needs a space.

Response: Thank you for your suggestion. We have revised this and highlighted in red on Page 10, Line 191.

8 . line 189, the sentence "This enhancement was..." should be placed in line 187.

Response: Thank you for your concern and comments. We have replaced the statement on Page 10, Line 197.

9 . line 193, provide full name for GST.

Response: Thank you for your valuable suggestions. We have rephrased this in the revised manuscript with Glutathione S-transferase (GST) on Page 12, Line 240.

10. line 198, inhibited instead of vanish; cite Figure 4C write after "failed to phosphorylate Hwa".

Response: Thank you for your suggestions. We have updated this in the revised manuscript on Page 10, Line 201.

11 . lines 204-209, could be explain more clearly.

Response: Thank you for your concern and suggestions. We have rephrased the statement for clarity in the revised manuscript on Page 11, Line 206-213.

12 . lines 289-290, a period is missing.

Response: Thank you for your suggestions. We have added a period and rephrased the statement in the revised manuscript.

13 . line 438, gifts from Anming Meng is an important author.

Response: Thank you for your helpful and valuable suggestions. We have removed this in the revised manuscript.

14 . Figure 2B explain that your using short and long exposures.

Response: Thank you for your concern and comments. What we wanted to show is the band intensities with different exposure times. For clarity, we performed additional experiments and have updated this figure with a single representative blot and statistics in the revised manuscript.

15 . Figure 4A reads pWha instead of pHwa.

Response: Thank you for your helpful suggestions. We have corrected this typo in the revised manuscript.

16 . Figure 6D, it would be better to separate to right panel (statistics) from the in-situ images.

Response: Thank you for your valuable comments and suggestions. We have reorganized the Figure 6 in the revised manuscript to separate the right panel (statistics) as **Fig. 6E**.

17 . Figure 7G, Hwa genotypes too small to read.

Response: Thank you for your constructive suggestions. We have reorganized the **Fig. 7** with bigger font and new data of S168A homo mutants in the revised manuscript.

Reviewer #3 (Remarks to the Author):

This manuscript provides compelling results showing the importance of phosphorylation of Serine 168 in the Huluwa (Hwa) protein, a new component of B-Catenin signaling that the authors previously showed acts as a dorsal determinant in axis formation in zebrafish and *Xenopus*. The authors new results reveal in loss and gain of function studies in zebrafish, as well as biochemical studies in HEK293 cells that Cdk16, Cdk2, and GSK3B can phosphorylate Hwa S168 and that this phosphorylation is critical for its dorsal axis induction function. They further show that Ccnyl potentiates Cdk16 activation, likely through its known role in phosphorylating and activating Cdk16. Interestingly, the Wnt receptor Lrp6 contains the same PPPSP motif that is phosphorylated by GSK3B. Lastly, they make a knock-in allele mutating S168 to alanine in the zebrafish, demonstrating that it is essential to axis formation, behaving the same as a null allele. This is exciting new work deciphering the regulatory mechanism of Hwa function, a novel protein key to dorsal axis formation. Prior to publication, several points below should be addressed.

Response: We really appreciate your positive assessment and evaluation of our work.

1. In vitro transcribed mRNA can have variable capping efficiencies, which can lead to variable activity from one batch of mRNA to the next, making it difficult to determine activity levels based on mRNA amount injected. For Hwa mutated mRNAs in Fig 1 that are deemed to have reduced activity, quantitative western blots should be performed to ensure the same amount of protein is made. Also some mutant Hwa forms could generate unstable proteins, rather than affecting activity perse.

Response: Thank you for your kind and valuable suggestions. Generally, we injected a group of mRNAs produced from the same batch to minimize variations introduced by human manipulation. In this study, we focused on Ser168, the mutation that was found to reduce the activity most dramatically. We then checked the expression of WT and S168A mRNAs in zebrafish embryos injected with the same dose from the same batch, using Flag-GFP mRNA as the internal control. Quantitative western blotting demonstrated that the amounts of expressed proteins were similar/comparable, indicating that the S168A mutation does not alter the stability of Hwa proteins, as shown in **Supplementary Fig. 1a-b**. The results from HEK293T cells

further confirmed that the S168A mutation did not affect protein stability, as shown in **Supplementary Fig. 1c-d**.

Supplementary Fig. 1. S168A point mutation did not affect the stability of Hwa.

(a) Immunoblotting of Hwa-HA and Flag-GFP in zebrafish embryos injected with 200 pg WT or S168A mutant hwa-HA mRNA with 200 pg Flag-gfp mRNA at the 1-cell stage and harvested at 4 hpf. (b) Quantifications of relative Hwa protein levels in embryos treated as in (a), N=4. (c) Immunoblotting of Hwa in cells transfected with wild-type or S168A Hwa-HA plasmids, followed by treatment with cycloheximide (CHX; 100 µg/ml) and harvesting at the indicated time points. (d) Quantifications of relative Hwa protein levels treated as in (c), N=4. GFP (a) and α-tubulin (c) were used as references for quantification. N, number of biological replicates; Significant differences are indicated by ns ≥ 0.05, *p < 0.05, **p < 0.01, ***p < 0.001, and ****p < 0.0001.

2. It seems that the Non-phosphorylated peptide also competes for the Hwa phospho-antibody in Fig S2A. Quantitation would clarify this.

Response: Thank you for your concern and suggestions. We have followed your suggestions and performed additional experiments with a purified pHwa antibody instead of the #11485 serum. We have thoroughly updated this figure with representative blots and statistics (**Supplementary Fig. 2**) to demonstrate the specificity of the pHwa antibody. We are sorry for the confusion and appreciate your understanding.

Supplementary Fig. 2. Verification of the specificity of the pHwa antibody. (a) Immunoblotting of pHwa from samples treated with λ -PPase or λ -PPase & phosphatase inhibitor. (b) Quantifications of relative pHwa levels treated as in (a), N=4. (c) Immunoblotting of pHwa in HEK293T cells transfected with wild-type or S168A mutant of Hwa-HA. (d) Quantifications of relative pHwa levels treated as in (c), N=4. (e) Immunoblotting of pHwa in HEK293T cells transfected with wild-type or mutants of Hwa-Flag. (f) Quantifications of relative pHwa treated as in (e), N=4. N, number of biological replicates; Significant differences are indicated by ns ≥ 0.05 , ***p < 0.001, and ****p < 0.0001, with individual p-values shown.

3. In S2B, what does weak and strong refer to, and the numbers above the strong blot? Assuming the numbers above the blot are quantitation (are they normalized to anti-Flag?), the phosphatase inhibitor seems modestly effective. To show significance in the difference, additional biological replicates and statistics would strengthen the point.

Response: Thank you for your concern and valuable suggestions. In the previous Fig. S2B, the "weak" and "strong" refer to "short time exposure with weak signals" or "long time exposure with strong signals", respectively. In addition, the "number" above the

blot indicates the average intensity of each band normalized to anti-Flag. To show significance with biological replicates, we have followed your suggestions, performed additional experiments and updated these with quantitative and statistical bar graphs, including the mean±SD and p-values, in the revised manuscript.

4. Why can CDK16 + Ccn1l rescue Hwa mutants to V1 in Fig 5D? This is surprising.

Response: Thank you for your concern and comments. In terms of the rescue effects of Cdk16/Ccn1l in *hwa* mutants, it is possible that this is due to the presence of very low levels of *hwa* RNAs/proteins in the *tsu01sm* mutants. However, it is clear that the rescue efficiency is very low, with only a tiny part of the embryos being partially restored to the V1 type. The presence of *hwa* mRNA in the *tsu01sm* mutants was confirmed by the RNA-seq results shown below (the purple block indicates the sequencing reads).

5. The increase in rescue by injecting Hwa + GSKB or *hwa* and *cdk2* is mild. Is the difference to Hwa injection alone in Fig 5E,F statistically significant? The modest rescue may reflect the presence of the endogenous activating kinase's in the embryo. An increase in *chordin* expression is clear though in Fig 5G.

Response: Thank you for your concern and comments. We are sorry for the confusion generated in the previous presentation. In the previous Fig.5E and 5F, the rescue injections were done at a later stage (at the 8-16-cell stage for *hwa*+*cdk2* and at the 16-32-cell stage for *hwa*+GSK3 β). In terms of the *chordin* detection shown in Fig. 5G, the mRNAs were injected at the 1-cell stage. Therefore, we performed additional rescue experiments with injections at the 1-cell stage, as with Cdk16/Ccn1l, and

observed a more significant increase (**Fig. 5h-i**). As you mentioned, interference of endogenous activating kinases in embryos does indeed exist (Gsk and dozens of Cdk are expressed during the early stage), and we thus injected lower doses of *hwa* mRNA to provide a more sensitive condition to monitor the additive effect of these two kinases (**Fig. 5j**).

Fig. 5h-j. Cdk2 and GSK3 β enhanced the axis-inducing activity of Hwa. (h) Rescue efficiency of 5 pg *hwa*-HA mRNA alone or together with Myc-*cdk2* injected at the 1-cell stage in *Mhwa*^{*tsu01sm/tsu01sm*} embryos, N=3. (i) Expression levels of *boz* and *chd* were quantified by RT-qPCR in embryos rescued by different mRNA combinations of *hwa*-HA and Myc-*cdk2* as in (h), N=3. (j) Rescue efficiency of lower dose (1.0 or 0.5 pg) of *hwa*-HA mRNA alone or together with Flag-*mGSK3 β* and Myc-*cdk2* injected at the 1-cell stage in *Mhwa*^{*tsu01sm/tsu01sm*} embryos, N=3. Phenotypes were grouped as in (h). *ef4g2a* was the internal reference in (g, i). V, ventralized; N, normal; D, dorsalized; V2<V1<N<D1<D2; N, number of biological replicates; n, total number of embryos in each treatment; Significant differences between treatments are indicated by ns \geq 0.05, * $p < 0.05$, ** $p < 0.01$, *** $p < 0.001$, and **** $p < 0.0001$.

6. Why does the *ccnyl* + 100 pg *cdk2*-DN attenuate axis formation equally or better to 200 pg *cdk2*-DN? Please clarify the logic behind adding the *ccnyl* here?

Response: Thank you for your concern and comments. Do you refer to that of Cdk16-DN? The D-to-N mutation is located in the catalytic domain of Cdk16 kinase to generate a dominant-negative form. As shown in **Supplementary Fig. 6e-f**, the K222R mutation (a kinase-inactivating mutation in the catalytic domain) enhanced the interaction between Hwa and Cdk16, which was further elevated by the presence of *Ccnyl1*. We suggest that *Ccnyl1* potentiated and increased the binding affinity of

Cdk16 to Hwa, regardless of the activity of Cdk16. Thus, we suppose that the presence of Ccn11 could promote the binding of Cdk16DN to Hwa and increase the inhibitory effect of Cdk16DN.

Supplementary Fig. 6e-f. Cdk16 interacted with Hwa. (e) Coimmunoprecipitation of Hwa-Flag with wild-type and K222R mutant Myc-Cdk16 in the absence or presence of Ccn11-HA. The kinase-inactivating form Cdk16 showed a stronger interaction with Hwa in the presence of Ccn11-HA. (f) Quantifications of relative Co-IP Hwa levels in HEK293T cells treated as in (e), N=3. Total Hwa proteins were used as loading controls for quantification. Significant differences are indicated by *p < 0.05 and ***p < 0.001, with individual p-values illustrated.

7. Excellent n values are provided for most experiments, however, N values should also be included for the number of biological replicates. At least 2 biological replicates should be performed, including for the western blots, and the number of such replicates should be included.

Response: Thank you for your comments and constructive suggestions. We have followed your advice and have provided the statistical results in the revised manuscript. For microinjections in zebrafish embryos, both the total number of embryos (n) and the number of biological replicates (N) have been illustrated in the figures and legends. For the western blots, both representative immunoblotting figures and statistical bar graphs (N≥3) have been displayed.

8. A typo in Fig 4A, pWha.

Response: Thank you for your concern and helpful suggestions. We have corrected the typo and have updated the data in **Supplementary Fig 5a & 5c**.

9. What stage are the embryos in Fig S5?

Response: Thank you for your valuable suggestions. The embryos were at the 1k-cell stage, which is generally used to detect Hwa/ β -catenin signal activation. We have added this information in the corresponding figure and figure legend in **Supplementary Fig. 6**.

Supplementary Fig. 6. Cdk16 was recruited to plasm membrane by Ccny. (a) Immunofluorescence of Cdk16-Myc and Ccny-HA in zebrafish embryos **at the 1k-cell stage**, with arrowheads indicating the plasm membrane located Cdk16-Myc. (b) Immunofluorescence of Cdk16-Myc and Ccny-HA in HEK293T cells. Cells were stained with DAPI in blue; Scale bar, 20 μ m.